



**Estimations of Global Shortwave Direct Aerosol Radiative Effects Above Opaque Water Clouds**
**Using a Combination of A-Train Satellite Sensors**
Meloë S. Kacenelenbogen [1]
Mark A. Vaughan [2]
Jens Redemann [3]
Stuart A. Young[4]
Zhaoyan Liu [2,4]
Yongxiang Hu[2]
Ali H. Omar [2]
Samuel LeBlanc [1],
Yohei Shinozuka [1],
John Livingston [1],
Qin Zhang [1],
Kathleen A. Powell [2]
[1]Bay Area Environmental Research Institute, Sonoma, CA, USA
[2]NASA Langley Research Center, Hampton, VA, USA
[3]University of Oklahoma, 120 David L. Boren Blvd., Suite 5900, Norman, OK
[4]Science Systems and Applications, Inc., Hampton, Virginia, USA
*Correspondence to:* Meloë S. Kacenelenbogen (meloe.s.kacenelenbogen@nasa.gov)





## Abstract

All-sky Direct Aerosol Radiative Effects (DARE) play a significant yet still uncertain role in climate. This is partly due to poorly quantified radiative properties of Aerosol Above Clouds (AAC). We compute global estimates of short-wave top-of-atmosphere DARE over Opaque Water Clouds (OWC), $DARE_{OWC}$, using observation-based aerosol and cloud radiative properties from a combination of A-Train satellite sensors and a radiative transfer model. There are three major differences between our $DARE_{OWC}$ calculations and previous studies: (1) we use the Depolarization Ratio method (DR) on CALIOP (Cloud Aerosol LIdar with Orthogonal Polarization) Level 1 measurements to compute the AAC frequencies of occurrence and the AAC Aerosol Optical Depths (AOD), thus introducing fewer uncertainties compared to using the CALIOP standard product; (2) we apply our calculations globally, instead of focusing exclusively on regional AAC "hotspots" such as the southeast Atlantic; and (3) instead of the traditional look-up table approach, we use a combination of satellite-based sensors to obtain AAC intensive radiative properties. Our results agree with previous findings on the dominant locations of AAC (South and North East Pacific, Tropical and South East Atlantic, northern Indian Ocean and North West Pacific), the season of maximum occurrence, aerosol optical depths (a majority in the 0.01-0.02 range and that can exceed 0.2 at 532 nm) and aerosol extinction-to-backscatter ratios (a majority in the 40-50 sr range at 532 nm which is typical of dust aerosols) over the globe. We find positive averages of global seasonal $DARE_{OWC}$ between 0.13 and 0.26 W·m$^{-2}$ (i.e., a warming effect on climate). Regional seasonal $DARE_{OWC}$ values range from -0.06 W·m$^{-2}$ in the Indian Ocean, offshore from western Australia (in March-April-May) to 2.87 W·m$^{-2}$ in the South East Atlantic (in September-





October-November). High positive values are usually paired with high aerosol optical depths (>0.1) and
low single scattering albedos (<0.94), representative of, e.g., biomass burning aerosols. Because we use
different spatial domains, temporal periods, satellite sensors, detection methods, and/or associated
uncertainties, the $DARE_{OWC}$ estimates in this study are not directly comparable to previous peer-
reviewed results. Despite these differences, we emphasize that the $DARE_{OWC}$ estimates derived in this
study are generally higher than previously reported. The primary reasons for our higher estimates are (i)
the possible underestimate of the number of dust-dominated AAC cases in our study; (ii) our use of
Level 1 CALIOP products (instead of CALIOP Level 2 products in previous studies) for the detection
and quantification of AAC aerosol optical depths, which leads to larger estimates of AOD above OWC;
and (iii) our use of gridded 4°x5° seasonal means of aerosol and cloud properties in our $DARE_{OWC}$
calculations instead of simultaneously derived aerosol and cloud properties from a combination of A-
Train satellite sensors. Each of these areas is explored in depth with detailed discussions that explain
both rationale for our specific approach and the subsequent ramifications for our DARE calculations.



## ACRONYMS

| | |
|---|---|
| AAC | Aerosol-Above-Clouds |
| AAOD | Absorption Aerosol Optical Depth |
| AOD | Aerosol Optical Depth |
| $\tau^{DR}_{AAC}$ | Aerosol Optical Depth above clouds using the DR method |
| AeroCom | Aerosol Comparisons between Observations and Models |
| AERONET | AErosol RObotic NETwork |
| AMSR-E | Advanced Microwave Scanning Radiometer - Earth Observing System |
| ARCTAS | Arctic Research of the Composition of the Troposphere from Aircraft and Satellites |
| ASR | integrated Attenuated Scattering Ratio |
| BRDF | Bidirectional Reflectance Distribution Function |
| CAC | Clear Air above Cloud |
| CALIOP | Cloud Aerosol LIdar with Orthogonal Polarization |
| CALIPSO | Cloud-Aerosol Lidar and Infrared Pathfinder Satellite Observations |
| CERES | Clouds and the Earth's Radiant Energy System |
| CF | Cloud Fraction |
| CloudSat | NASA Earth observation satellite |
| COD | Cloud Optical Depth |





| CR | Color Ratio technique |
|---|---|
| $DARE_{all\text{-}sky}$ | Direct Aerosol Radiative Effect in all-sky conditions (cloudy and non-cloudy) |
| $DARE_{cloudy}$ | Direct Aerosol Radiative Effect in cloudy conditions |
| $DARE_{non\text{-}cloudy}$ | Direct Aerosol Radiative Effect in non-cloudy conditions (clear-skies) |
| $DARE_{OWC}$ | Direct Aerosol Radiative Effect above opaque water clouds |
| DISORT | DIScrete ORdinate Radiative Transfer solvers |
| DR | Depolarization Ratio technique |
| $\delta^{OWC}$ | layer-integrated volume depolarization ratio |
| $f_{AAC}$ | AAC frequency of occurrence |
| HSRL | High Spectral Resolution Lidar |
| IAB | Integrated Attenuated Backscatter |
| IBS | Integrated aerosol Backscatter |
| InWA | Indian ocean, offshore from West Australia |
| LUT | Look Up Table |
| LWP | Liquid Water Path |
| MBL | Marine Boundary Layer |
| MCD43GF | MODIS BRDF/Albedo/NBAR CMG Gap-Filled Products |
| MODIS | MODerate Imaging Spectroradiometer |
| $\eta^{OWC}$ | layer effective multiple scattering factor |



| | |
|---|---|
| NEAs | North East Asia |
| NEPa | North East Pacific ocean |
| NWPa | North West Pacific ocean |
| OMI | Ozone Monitoring Instrument |
| ORACLES | ObseRvations of Aerosols above CLouds and their intEractionS |
| OWC | Opaque Water Cloud |
| POLDER | Polarization and Directionality of Earth's Reflectances |
| PBL | Planetary Boundary Layer |
| $R_e$ | Cloud droplet effective radius |
| RT | Radiative Transfer scheme |
| SAA | South Atlantic Anomaly |
| $S_a$ | Aerosol extinction-to-backscatter (lidar) ratio |
| $S_{AAC}$ | Aerosol extinction-to-backscatter (lidar) ratio above clouds |
| $S_c$ | Cloud extinction-to-backscatter (lidar) ratio |
| SCIAMACHY | Scanning Imaging Absorption Spectrometer for Atmospheric Cartography |
| SEAs | South East Asia |
| SEAt | South East Atlantic ocean |
| SEPa | South East Pacific ocean |
| SEVIRI | Spinning Enhanced Visible and InfraRed Imager |
| SNR | Signal-to-Noise Ratio |



| SS | Single Scattering |
| SSA | Single Scattering Albedo |
| SW | Short Wave |
| TAt | Tropical Atlantic ocean |
| TOA | Top Of Atmosphere |



## 1. Introduction

The Direct Aerosol Radiative Effect (DARE) is defined as the change in the upwelling radiative flux ($F^\uparrow$) at the top of the atmosphere (TOA) due to aerosols. Measured values of DARE depend on the accuracy and the geometry of the observation(s), the concentrations of various atmospheric constituents (e.g., aerosols, clouds, and atmospheric gases) and their radiative properties, and the Earth's surface reflectance. All-sky DARE ($DARE_{all-sky}$) combines contributions from DARE under cloudy conditions ($DARE_{cloudy}$) and DARE under cloud-free conditions ($DARE_{non-cloudy}$):

$$DARE_{all-sky} = DARE_{cloudy} \times \text{Cloud Fraction} + DARE_{non-cloudy} \times (1 - \text{Cloud Fraction}) \qquad \text{Eq. (1)}$$

According to Yu et al., [2006], substantial progress has been made in the assessment of $DARE_{non-cloudy}$ using satellite and in situ data. Further evidence is provided in a companion to our study, Redemann et al. [2018], which use A-Train aerosol observations to constrain $DARE_{non-cloudy}$ and compares the results with AeroCom (Aerosol Comparisons between Observations and Models) results (see Appendix A for further details). However, traditional passive aerosol remote sensing techniques are limited only to clear-sky conditions and significant efforts are required to estimate $DARE_{cloudy}$. Moreover, simulations of $DARE_{cloudy}$ from various AeroCom models in Schulz et al. [2006] (see their figure 6) show large disparities. Our study focuses on Aerosol Above Cloud (AAC) scenes over the globe and subsequent estimates of $DARE_{cloudy}$ (i.e., the instantaneous short wave (SW) upwelling TOA reflected radiative fluxes due to clouds only minus SW upwelling TOA fluxes due to clouds with overlying aerosols). Let us note that, ideally, TOA $DARE_{cloudy}$ should include aerosols below, in-between and above clouds. Here we assume that TOA $DARE_{cloudy}$ is only caused by aerosols above clouds. Table 1 lists TOA SW



DARE$_{cloudy}$ results that use satellite observations in the literature, together with assumptions in their
calculations. Compared to the peer-reviewed studies of Table 1, our study marks a departure on three
accounts. First, most peer-reviewed DARE$_{cloudy}$ calculations focus primarily on the South East Atlantic
(SEAt e.g., [Chand et al., 2009, Wilcox et al., 2012, Peters et al., 2011, De Graaf et al., 2012, 2014,
Meyer et al., 2013, 2015, Peers et al., 2015, Feng and Christopher, 2015] in Table 1). Second, our
results use a combination of A-Train satellite sensors (i.e., MODIS-OMI-CALIOP), instead of the
Look-Up-Table (LUT) approach used in the other studies of Table 1, to obtain estimates of the intensive
aerosol radiative properties above clouds. Third, the peer-reviewed global DARE$_{cloudy}$ calculations in
Table 1 use standard products from the active satellite sensor Cloud Aerosol LIdar with Orthogonal
Polarization (CALIOP) for either AAC Aerosol Optical Depth (AOD) and/or aerosol and cloud vertical
distribution information in the atmosphere [Zhang et al., 2014, 2016, Matus et al., 2015, Oikawa et al.,
2013]. In our case, we estimate DARE$_{cloudy}$ globally by using an alternate method applied to CALIOP
Level 1 measurements [Hu et al., 2007b; Chand et al., 2008; Liu et al., 2015] to obtain AAC AOD and
the AAC frequency of occurrence. In the sections below, we explain why we have used such a method,
instead of other passive or active satellite sensor techniques.
**Table 1:** TOA SW DARE$_{cloudy}$ calculations that use satellite observations in the literature and specific
assumptions in the calculations. See also the theoretical study by Chang and Christopher et al. [2017]
(i.e. they impose fixed COD, Re, AOD, aerosol radiative properties, and aerosol / cloud vertical
distribution) and the study by Costantino and Bréon et al. [2013] (their method uses MODIS-derived
cloud microphysics that are not corrected for overlying aerosols). When not specified, the study uses the
standard CALIOP data product; otherwise, it uses the DR (Depolarization Ratio) or CR (Color Ratio)



technique on CALIOP measurements. $MODIS^A$ and $MODIS^T$ respectively denote the AQUA or
TERRA platform. SEAt: South East Atlantic. LUT: Look Up Table. See acronyms for satellite sensors
MODIS, CALIOP, CloudSat, POLDER, CERES and AMSR-E.

| Reference | Domain | Satellite sensor(s) used for DARE$_{cloudy}$ calculations | | | |
|---|---|---|---|---|---|
| | | Cloud properties (e.g. COD, albedo, fraction) | AOD | Aerosol radiative properties (e.g. SSA, g) | Vertical distribution of aerosol and cloud |
| Chand et al. [2009] | SEAt | $MODIS^T$ | $CALIOP^{CR}$ | Fixed value | Assumed constant |
| Wilcox [2012] | SEAt | $MODIS^A$, AMSR-E | CERES provides upwelling shortwave flux | | |
| Peters et al. [2011] | Atlantic | $MODIS^A$, AMSR-E | CERES provides upwelling shortwave flux | | |
| De Graaf et al. [2012, 2014] | SEAt | Direct determination of DARE$_{cloudy}$ by building LUT of cloud and aerosol-free reflectances | | | |
| Meyer et al. [2013] | SEAt | $MODIS^A$ | CALIOP | LUT approach | CALIOP |
| Zhang et al. [2014, 2016] | Globe | $MODIS^A$, CALIOP (uses probability density function of CALIOP above-cloud AOD and underlying MODIS COD) | | LUT approach | CALIOP |
| Meyer et al. [2015] | SEAt | $MODIS^A$ (simultaneous retrieval of above-cloud AOD, COD and $R_e$) | | LUT approach | Assumed constant |
| Peers et al. [2015] | SEAt | POLDER (simultaneous retrieval of above-cloud aerosol OD, size and single scattering albedo, cloud optical depth and cloud top height) | | | |
| Feng and Christopher [2015] | SEAt | $MODIS^A$, CERES | CERES provides upwelling shortwave flux | | |



| Reference | Domain | Satellite sensor(s) used for DARE$_{cloudy}$ calculations | | | |
|---|---|---|---|---|---|
| | | Cloud properties (e.g. COD, albedo, fraction) | AOD | Aerosol radiative properties (e.g. SSA, g) | Vertical distribution of aerosol and cloud |
| Matus et al. [2015] | Globe | CloudSat, MODIS$^A$, CALIOP | CALIOP | LUT approach | CloudSat, CALIOP |
| Oikawa et al. [2013] | Globe | CALIOP, MODIS$^A$ | CALIOP | LUT approach | CALIOP |
| This study | Globe | MODIS$^A$ | CALIOP$^{DR}$ | MODIS$^A$, OMI, CALIOP | Assumed constant |


Table 2 lists some passive (i.e., Spinning Enhanced Visible and InfraRed Imager, SEVIRI, Moderate
Resolution Imaging Spectroradiometer, MODIS, Polarization and Directionality of Earth's
Reflectances, POLDER, Ozone Monitoring Instrument, OMI or the Scanning Imaging Absorption
Spectrometer for Atmospheric Chartography, SCIAMACHY) and active (i.e., CALIOP and CloudSat)
satellite sensors that were used to detect and quantify the AAC AODs. Among the peer-reviewed
studies of Table 2, those few that present DARE$_{cloudy}$ results (see Table 1) are denoted by a "+" sign in
the first column.
**Table 2:** Studies that observe AAC using passive and active satellite sensors (i.e., from left to right,
SEVIRI, POLDER, CloudSat, OMI, MODIS, SCIAMACHY, CALIOP; see acronyms). When using
CALIOP, the authors either use the standard Level 2 products (Std), the Depolarization method (DR)
[Hu et al., 2007b] or the color ratio method (CR) [Chand et al., 2008]. SEAt stands for SE Atlantic,




SEAs for SE Asia, NEAs for NE Asia and TAt for Tropical Atlantic. The "+" sign in the first column
denotes the presence of DARE$_{cloudy}$ calculations.

| | Reference | Domain | \multicolumn{7}{c}{Satellite sensor(s) used for aerosol-above-cloud detection} | | | | | | |
| --- | --- | --- | --- | --- | --- | --- | --- | --- | --- |
| | | | SEVIRI | POLDER | CloudS | OMI | MODIS | SCIAMA | CALIOP |
| 1 | Chang and Christopher [2016, 2017+] | SEAt | ■ | | | | ■ | | |
| 2 | Waquet et al. [2013a] | Globe | | ■ | | | | | |
| 3 | Waquet et al. [2009, 2013b] | SEAt, TAt | | ■ | | | | | |
| 4 | Peers et al. [2015]+ | SEAt | | ■ | | | | | |
| 5 | Jethva et al [2013, 2014] | SEAt, TAt | | | | | ■ | | |
| 6 | Torres et al. [2012] | SEAt | | | | ■ | | | |
| 7 | Peters et al. [2011]+ | Atlantic | | | | ■ | | | |
| 8 | De Graaf et al. [2012, 2014]+ | SEAt | | | | | | ■ | |
| 9 | Meyer et al. [2015]+ | SEAt | | | | | ■ | | |
| 10 | Feng and Christopher [2015]+ | SEAt | | | | ■ | | | |
| 11 | Sayer et al. [2016] | SEAt, SEAs | | | | | ■ | | |
| 12 | Matus et al. [2015]+ | Globe | | | ■ | | | | Std |
| 13 | Alfaro-Contreras et al. [2016] | Globe | | | | ■ | ■ | | Std |
| 14 | Alfaro-Contreras et al. [2014] | SEAt, SEAs | | | | ■ | ■ | | Std |
| 15 | Devasthale and Thomas [2011] | Globe | | | | | | | Std |
| 16 | Yu et al. [2012] | SEAt, TAt | | | | ■ | | | Std |
| 17 | Wilcox [2012]+ | SEAt | | | | ■ | | | Std |
| 18 | Meyer et al. [2013]+ | SEAt | | | | | | | Std |
| 19 | Zhang et al. [2014, 2016]+ | Globe | | | | | | | Std |
| 20 | Oikawa et al. [2013]+ | Globe | | | | | ■ | | Std |
| 21 | Chung et al. [2016] | Globe | | | | | | | Std |
| 22 | Chand et al. [2008] | SEAt | | | | | | | CR, DR |



| | Reference | Domain | Satellite sensor(s) used for aerosol-above-cloud detection | | | | | | |
|---|---|---|---|---|---|---|---|---|---|
| | | | SEVIRI | POLDER | CloudS | OMI | MODIS | SCIAMA | CALIOP |
| 23 | Chand et al. [2009][+] | SEAt | | | | | | | CR |
| 24 | Deaconu et al. [2017] | Globe | | | | | | | Std, DR |
| 25 | Liu et al. [2015] | SEAt, TAt | | | | | | | DR |
| 26 | This study[+] | Globe | | | | | | | DR |

The brightening of clear patches near clouds [Wen et al., 2007] (i.e., "3-D cloud radiative effect" or
"cloud adjacency effect") can introduce biases into the current passive satellite AAC retrieval
techniques (i.e., lines 1-11 of Table 2). To minimize these biases, this study relies primarily on CALIOP
observations [Winker et al., 2009]. CALIOP is a three-channel elastic backscatter lidar with a narrow
field of view and a narrow source of illuminating radiation, which limits cloud adjacency effects and the
subsequent cloud contamination of aerosol data products [Zhang et al., 2005; Wen et al., 2007; Várnai
and Marshak, 2009]. CALIOP measures high-resolution (1/3 km in the horizontal and 30m in the
vertical in low and middle troposphere) profiles of the attenuated backscatter from aerosols and clouds
at visible (532 nm) and near-infrared (1064 nm) wavelengths along with polarized backscatter in the
visible channel [Hunt et al., 2009]. These data are distributed as part of the Level 1 CALIOP products.
The Level 2 products are derived from the Level 1 products using a succession of sophisticated retrieval
algorithms [Winker et al., 2009]. The Level 2 processing is composed of a feature detection scheme
[Vaughan et al., 2009], a module that classifies features according to layer type (i.e., cloud versus
aerosol) [Liu et al., 2010] and subtype (i.e., aerosol species) [Omar et al., 2009], and, finally, an
extinction retrieval algorithm [Young and Vaughan, 2009] that retrieves profiles of aerosol backscatter
and extinction coefficients and the total column AOD based on modeled values of the extinction-to-



backscatter ratio (also called lidar ratio and represented by the symbol $S_a$) inferred for each detected
aerosol layer subtype.
A few studies use standard CALIOP Level 2 Aerosol and Cloud Layer products to determine AAC
occurrence over the globe (see line 12-21 in Table 2). However, a study by Kacenelenbogen et al.
[2014] demonstrates that the standard version 3 CALIOP aerosol products substantially underreport the
occurrence frequency of AAC when aerosol optical depths are less than ~0.02, mostly because these
tenuous aerosol layers have attenuated backscatter coefficients less than the CALIOP detection
threshold. CALIOP's standard extinction (and optical depth) data products are only retrieved between
the tops and bases of detected features, and these boundaries may significantly underestimate the full
vertical extent of the layer (Kim et al., 2017; Thorsen et al., 2017; Toth et al., 2018). Furthermore, the
Kacenelenbogen et al. [2014] study found essentially no correlation between AAC AOD results
reported by the CALIOP and collocated NASA Langley airborne High Spectral Resolution Lidar
(HSRL). A subsequent study by Liu et al. [2015] shows that the CALIOP Level 2 standard aerosol data
products underestimate dust AAC AOD by ~26% over the Tropical Atlantic and smoke AAC AOD by
~39% over the SE Atlantic.
For these reasons, a few studies in Table 2 (see line 22-26) use alternate methods on Level 1 CALIOP
products, such as the Color Ratio (CR) [Chand et al., 2008] or the Depolarization Ratio (DR) [Hu et al.,
2007b; Liu et al., 2015] methods, instead of using the AOD reported in the CALIOP standard Level 2
products.



In this study, we use the DR method and a combination of CALIOP Level 1 and Level 2 data products
to compute global estimates of the AAC frequency of occurrence (i.e., $f_{AAC}$), the AAC AOD (i.e.,
$\tau^{DR}_{AAC}$) and the AAC extinction-to-backscatter ratios (i.e., $S_{AAC}$) (section 2.1). We then use CALIOP
results of $f_{AAC}$, $\tau^{DR}_{AAC}$ and other A-Train satellite products to compute global $DARE_{cloudy}$ (section 2.2).
Section 3 describes the geographical and seasonal distribution of global $f_{AAC}$ (section 3.1), $\tau^{DR}_{AAC}$ and
$S_{AAC}$ (section 3.2) and $DARE_{cloudy}$ results (section 3.3). Section 4 revisits some of the limitations in the
method and proposes ways to improve on these $DARE_{cloudy}$ calculations.
**2.   Method**

62       **2.1.       AAC optical depth and extinction-to-backscatter**

The DR method can also be called the "constrained opaque water cloud method" [Liu et al, 2015] as it
uses Opaque Water Clouds (OWCs) as reflectivity targets. The OWCs in this study are selected using
the five criteria listed in Table B2 of the appendix. Most importantly, (1) only one cloud can be detected
within a 5 km (15 shot) along-track average (which means, for example, that marine stratus below thin
cirrus are excluded). Furthermore, this one cloud must be (2) opaque (which means that low but
transparent clouds such as the ones reported in Leahy et al. [2012] are excluded), (3) spatially uniform
(i.e., detected at single-shot resolution within every laser pulse included in the 5 km averaging interval),
(4) assigned a high confidence score by the CALIOP cloud-aerosol discrimination (CAD) algorithm and
(5) identified as a high confidence water cloud by the CALIOP cloud phase identification
algorithm.When there is aerosol above OWCs, the lidar backscatter signal received from the underlying
water cloud is reduced in direct proportion to the two-way transmittance of the aerosol layer above.



Based on Hu et al. [2007a, 2007b], Eq. (2) describes how we compute $\tau^{DR}_{AAC}$ using the DR method
above OWCs.
$\tau^{DR}_{AAC} = -0.5 \times \ln[IAB^{OWC}_{SS,AAC} / IAB^{OWC}_{SS,CAC}]$                                   Eq. (2)
Here $IAB^{OWC}_{SS,AAC}$ is the single scattering value (subscript SS) of the layer-integrated attenuated
backscatter (IAB) for an OWC underlying one or more aerosol layer(s) above the cloud. $IAB^{OWC}_{SS,CAC}$
is the single scattering value of the IAB for an OWC underlying Clear air Above Cloud (CAC). By
CAC, we mean that there are no aerosols detected above the OWC. In this study, we consider $\tau^{DR}_{AAC}$
valid when positive. According to Eq. (2), this means that $IAB^{OWC}_{SS,AAC}$ needs to always be smaller in
magnitude than $IAB^{OWC}_{SS,CAC}$ and $\tau^{DR}_{AAC}$ equals zero when $IAB^{OWC}_{SS,AAC}$ equals $IAB^{OWC}_{SS,CAC}$.
Section B of the appendix provides additional information about the application of Eq. (2) and the
various steps needed to derive $\tau^{DR}_{AAC}$. We list the selection criteria used to identify the OWC dataset in
this study and describe the corrections required to obtain single-scattering estimates of IAB from
measurements that contain substantial contributions from multiple scattering (B1). We also describe the
technique used for distinguishing between CAC and AAC conditions (B2), and illustrate our derivation
of an empirical parameterization of $IAB^{OWC}_{SS,CAC}$ as a global function of latitude and longitude (B3).
As reported in Table 2, the CALIOP DR method was used to study the African dust transport pathway
over the Tropical Atlantic [Liu et al., 2015] and the African smoke transport pathway over the South
East Atlantic [Liu et al., 2015; Chand et al., 2008, 2009]. More recently, the CALIOP DR method was
also used by Deaconu et al. [2017] to assess POLDER AAC AOD values [Waquet et al., 2009, 2013b
and Peers et al., 2015] over the globe. In this study, we extend the previous regional studies of [Liu et



al., 2015 and Chand et al., 2008, 2009] to derive global CALIOP-based AAC AOD estimates. $S_{AAC}$
values are then computed by solving Eq. (15) of Fernald et al. [1972], constrained by valid (i.e.,
positive) $\tau^{DR}_{AAC}$ and using the GEOS-5 molecular and ozone number density values and the CALIOP
Level 1 attenuated backscatter profiles (see step S5 in Table B1). Let us note that, in our study, the
ability to retrieve CALIOP $S_{AAC}$ has no bearing on the accuracy of our CALIOP $\tau^{DR}_{AAC}$ retrievals. The
accuracy of $\tau^{DR}_{AAC}$ depends on measurements of targets of very high signal-to-noise ratio (SNR) such
as OWCs in clear skies and OWCs underlying aerosols layers. On the other hand, many $S_{AAC}$ retrievals
depend on very low SNR measurements obtained from the weakly scattering and vertically diffuse
aerosol layers above OWCs.

### 2.2.    AAC Direct Aerosol Radiative Effects

Having first retrieved global values of $\tau^{DR}_{AAC}$ from the CALIOP measurements, we then compute
global estimates of $DARE_{cloudy}$ using DISORT (DIScrete ORdinate Radiative Transfer; Stamnes et al.,
1988, Buras et al., 2011), a six-stream plane-parallel radiative transfer model with molecular absorption
characterized by a correlated-k scheme [Fu and Liou, 1992] that is embedded within the LibRadtran
Radiative Transfer (RT) package [Emde et al., 2016]. Hereafter, our seasonally and spatially gridded (4º
x 5º) averaged shortwave (SW) (250 nm to 5600 nm) global TOA $DARE_{cloudy}$ results will be called
$DARE_{OWC}$, as they pertain to a specific category of clouds (i.e., OWCs) defined according to the
CALIOP data selection criteria set forth in Table B2. We list the following input parameters to DISORT
in order to derive estimates of $DARE_{OWC}$:





(1) **Atmospheric profiles** of pressure, temperature, air density, ozone, water vapor, $CO_2$, and $NO_2$
use standard US atmosphere profiles [Anderson et al., 1986].
(2) **Aerosol intensive radiative properties** (i.e. properties that depend solely on aerosol species,
and are unrelated to the aerosol amount) are informed by seasonal maps (4º x 5º, daytime in 2007)
of combined MODIS-OMI-CALIOP (MOC) retrieved median spectral extinction coefficients,
single scattering albedos and asymmetry parameters at 30 different wavelengths. As an example,
Figure A1 in the appendix shows the seasonal maps of MOC SSA at 546.3 nm that were used in the
calculation of $DARE_{OWC}$. These MOC retrievals, described in section A of the appendix, are at the
basis of a companion study [Redemann et al., 2018]. Let us note that we only use the shape of the
MOC extinction coefficient spectra and not its actual magnitude; the MOC spectral extinction
coefficient spectra is normalized to the seasonal 2008-2012 average value of either $\tau^{DR}_{AAC}$ or $\tau^{DR}_{AAC}$
x $f_{AAC}$ within each grid cell. Our method assumes similar aerosol radiative properties above clouds
and in near-by clear-sky regions.
(3) **Aerosol extensive radiative properties** (i.e., properties that depend on the aerosol amount
present in the atmosphere) are informed by seasonal maps (4º x 5º, nighttime from 2008 to 2012) of
either CALIOP $\tau^{DR}_{AAC}$ (see Eq. 2) or CALIOP $\tau^{DR}_{AAC}$ x $f_{AAC}$. We chose to use nighttime CALIOP
$\tau^{DR}_{AAC}$ or $\tau^{DR}_{AAC}$ x $f_{AAC}$ results in the estimation of $DARE_{OWC}$ because, at nighttime, the CALIOP
signal-to-noise-ratio (SNR) is not affected by ambient solar background and leads to a more
accurate measurement of the aerosol signal (compared to daytime). By doing this, we implicitly
chose a better accuracy in the aerosol extensive radiative properties over a temporal overlap
between aerosol extensive (nighttime) and intensive (daytime) radiative properties.



(4) **Cloud albedos** are computed from cloud droplet effective radius ($R_e$) and Cloud Optical Depth
(COD) information inferred from MODIS averaged monthly 1°x1° grids (i.e. liquid water cloud
products of MYD08_M3: "Cloud Effective Radius Liquid Mean Mean" and "Cloud Optical
Thickness Liquid Mean Mean" [Platnick et al. 2015]) from 2008 to 2012 (see Equations 1-9 of Peng
et al. [2002]). These maps are then further gridded (to 4°x5°) and seasonally averaged to match the
format of the aerosol radiative properties. Appendix figure A2 shows the seasonal maps of MODIS
COD that were used in the calculation of $DARE_{OWC}$.
(5) **Aerosol and cloud layer heights** are assumed constant over the globe (respectively between 3-
4km and 2-3km in this study), similar to other studies in Table 1 (e.g., Meyer et al. [2015]).
(6) **Earth's surface albedo** uses global gap-filled Terra and Aqua combined MODIS BRDF/albedo
products. It uses the 16-day closest product (i.e., MCD43GF) to the middle of each season (i.e., Jan
15th for DJF, April 15th for MAM, July 15th for JJA and October 15th for SON). In the open ocean,
the Cox and Munk [1954] sea surface albedo parameterization is applied with a wind speed of 10
ms$^{-1}$.
Using these inputs, **Daily $DARE_{OWC}$** results for each of the 4° x 5° grid cells are obtained by averaging
24 LibRadtran RT calculations, corresponding to 24 different sun positions at each hour of the day.
**3. Results**
**3.1.    AAC Occurrence Frequencies**
To provide the necessary context for interpreting our TOA radiative transfer calculations, we first
establish the observational AAC occurrence frequencies from which we will subsequently compute



estimates of DARE$_{OWC}$. Figure 1 illustrates the annual gridded mean (5 years) global occurrence
frequencies of a) single layer clouds, b) opaque water clouds that are suitable for the DR method and c)
aerosol-above-clouds cases using the DR method. Figure 1d) shows the difference between the number
of AAC cases using the DR method (i.e., number of cases with $\tau^{DR}_{AAC} > 0$) and the number of AAC
cases using the standard Version 3 CALIOP product.

a) **max: 76% median: 48% mean: 47%**

**Uniform single layer clouds
as % of 5km sample**

b) **max: 67% median: 4% mean: 7%**

**Opaque water cloud for DR method
as % of 5km sample**

c) **max: 42% median: 4% mean: 5%**

**Occurrence of $\tau^{DR}_{AAC} > 0$
as % of 5km sample**

d) **max: 36% min: -24% median: -1% mean: -1%**

**Difference in occurrence of $\tau^{DR}_{AAC} > 0$
and $\tau^{STD}_{AAC} > 0$
as % of 5km sample**



**Figure 1:** During nighttime, from 2008 to 2012 on a 4°x5°-grid: Occurrence frequencies of (a) uniform
single layer clouds (C1-C3 of Table B2), (b) opaque water clouds suitable for the DR method (C1-C5 of
Table B2; these clouds can be obstructed or unobstructed) and (c) AAC cases that show a positive
$\tau^{DR}_{AAC}$ at 532 nm. (d) shows the difference between the number of AAC cases using the DR method
(i.e., number of cases with $\tau^{DR}_{AAC} > 0$) and the number of AAC cases using the standard Version 3
CALIOP product (i.e., number of cases with $\tau^{STD}_{AAC} > 0$); CALIOP AAC cases using the standard
algorithm are defined as 5 km-columns showing an uppermost layer classified as aerosols and a cloud
layer anywhere below that aerosol layer; the cloud itself does not have to satisfy any of the criteria of
Table B2. Grid cells are 4° x 5° latitude/ longitude. The percentages in (a)-(d) use the number of 5 km
CALIOP samples within each grid cell as a reference. White pixels show either no CALIOP
observations, no CALIOP OWC detection, a small number of CALIOP unobstructed OWCs or a small
number of positive $\tau^{DR}_{AAC}$ values. The title of each map shows the global maximum, median and mean
values.
Uniform single layer clouds (i.e. C1-C3 of Table B2) are detected in ~47% of all 5 km CALIOP
samples over the globe (see Figure 1(a)). In other words, at any one time, approximately half of the
globe is covered by uniform single layer clouds. As expected, the highest occurrence of those clouds is
in the high and low latitude bands and especially over the southern oceans. According to Figure 1(b),
OWCs suitable for the DR method (i.e. C1-C5 of Table B2) are mostly in the marine stratocumulus
regions and represent a mean of 7% of all 5 km CALIOP samples over the globe. This significant



reduction from half-the-globe coverage is explained by the five criteria used to select OWCs for the
application of the DR method (i.e., C1-C5 of Table B2). The highest occurrence of OWCs can be found
offshore from the west coasts of North and South America, southwest Africa and Australia. In
particular, OWC cover ranges from 60 to 75 % over the region of SE Atlantic in August [Klein and
Hartmann, 1993]. Also, the southeastern Pacific region off the Peruvian and Chilean coasts is the
location of the largest and most persistent stratocumulus deck in the world [Klein and Hartmann, 1993].
The percentage of AAC cases (i.e., AAC cases showing positive $\tau^{DR}_{AAC}$) at the basis of our study is
very small compared to the total number of 5 km CALIOP profiles per grid cell (i.e. mean of 5% on
Figure 1(c)). This is primarily due to a small number of low OWC used for the DR method over the
globe (when comparing Figure 1(a) and 1(b)).
Figure 1(d) illustrates the difference in occurrence frequencies of AAC cases using the DR method
compared to the standard Version 3 CALIOP product; negative (positive) values in blue (red) show the
number of AAC cases that are missed (gained) by the DR method compared to using the standard
CALIOP products. Unlike Figure 1(c), the AAC cases in Figure 1(d) that use the CALIOP standard
product do not require any assumptions on the nature of the underlying cloud. Figure 1(d) shows that
we could be missing (in blue) AAC cases over most of the land surfaces and over the Arabian Sea, the
Tropical Atlantic and the SE Atlantic regions by using the DR method instead of the standard CALIOP
product. One reason for the lack of AAC cases offshore from the west coast of Africa in our dataset is
the filtering out of "unobstructed" but potentially aerosol-contaminated OWCs (see section B3 in the
appendix for more details). However, some regions such as the NE and SE Pacific exhibit up to 40%
more (in red) AAC cases when using the DR method. The SE Pacific region, especially offshore from



Chile, shows particularly tenuous aerosols, with attenuated backscatter values that typically fall below
the CALIOP detection limit and, hence, hampers the detection of AAC using the standard CALIOP
algorithm [Kacenelenbogen et al., 2014].
In the rest of this study, the frequency of occurrence of AAC, $f_{AAC}$, is defined as:
$f_{AAC} = N_{AAC}/N_{OWC}$                                                                 Eq. (3)
where $N_{AAC}$ is the number of AAC cases (i.e., cases showing a positive $\tau^{DR}_{AAC}$ at 532nm) and $N_{OWC}$ is
the number of OWCs within each 4º×5º grid cell. Let us note that different studies use different
references when computing the frequency of occurrence of AAC. The definition in Eq. (3) is similar to
the one in Zhang et al. [2016] (see their Eq. (1)) and different from Devasthale and Thomas [2011],
where $f_{AAC}$ is defined as the ratio of AAC cases to the total number of CALIOP observations (similar to
what is shown on Fig. 1(c)).
Figure 2 illustrates the global seasonal $f_{AAC}$ (see Eq. 3) from 2008 to 2012. We find a median global
$f_{AAC}$ of 58% to 61% with regional values that can reach more than 80% in some regions such as the SE
Atlantic, especially during the JJA season. The AAC occurrence frequencies in Fig. 2 generally agree
with previous findings [Zhang et al., 2016; Devasthale and Thomas, 2011] on the location and season of
highest $f_{AAC}$.



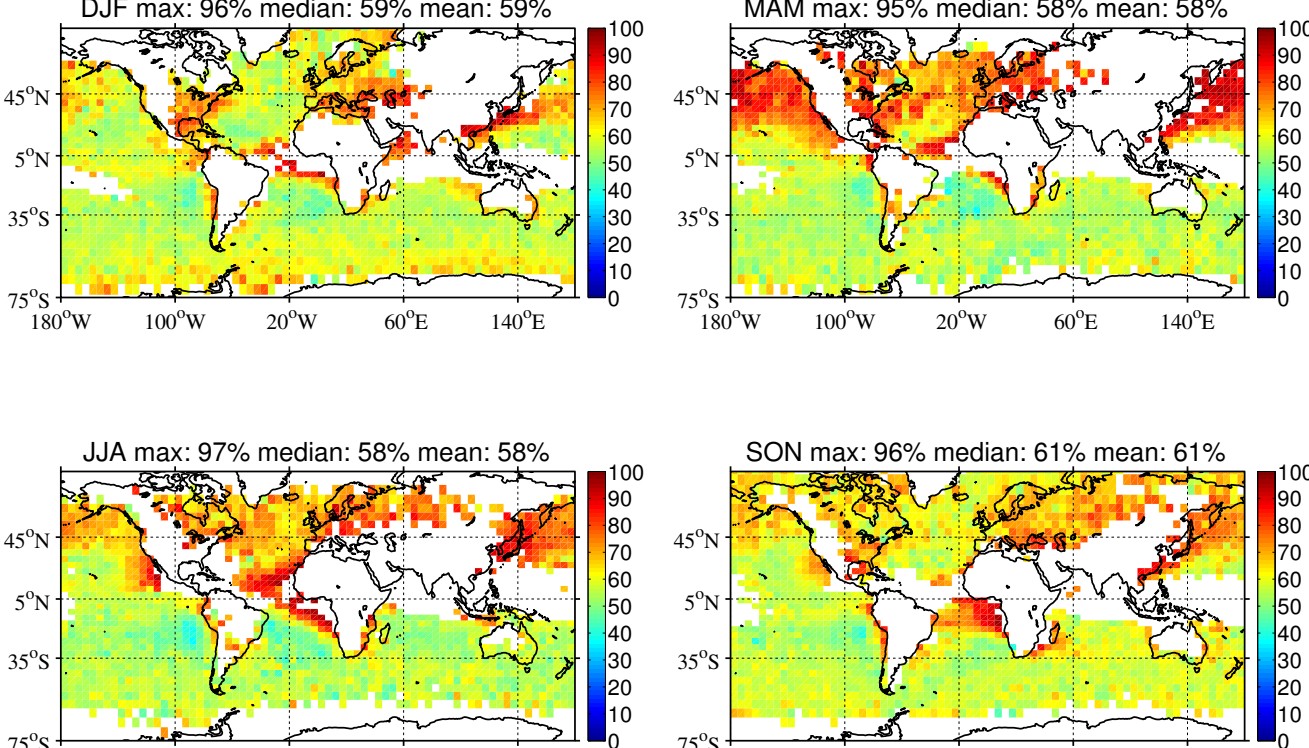

**Figure 2:** Global seasonal 4°x5° nighttime AAC occurrence frequency (noted $f_{AAC}$, see Eq. (3)) from 2008 to 2012. White pixels show either no CALIOP observations, a limited number of CALIOP unobstructed OWCs or a limited number of positive $\tau^{DR}_{AAC}$ values. White pixels are not considered in the global mean and median $f_{AAC}$ values in the title of each map. The title of each map shows the global maximum, median and mean values.

### 3.2. AAC Optical Depths, Extinction-to-Backscatter Ratios and South Atlantic Anomaly Effects

### 29    3.2.1 AAC Optical Depths

Figure 3 introduces the global, nighttime and multi-year (2008-2012) AAC optical depths ($\tau^{DR}_{AAC}$, see
Eq. 2) dataset that was computed in this study.

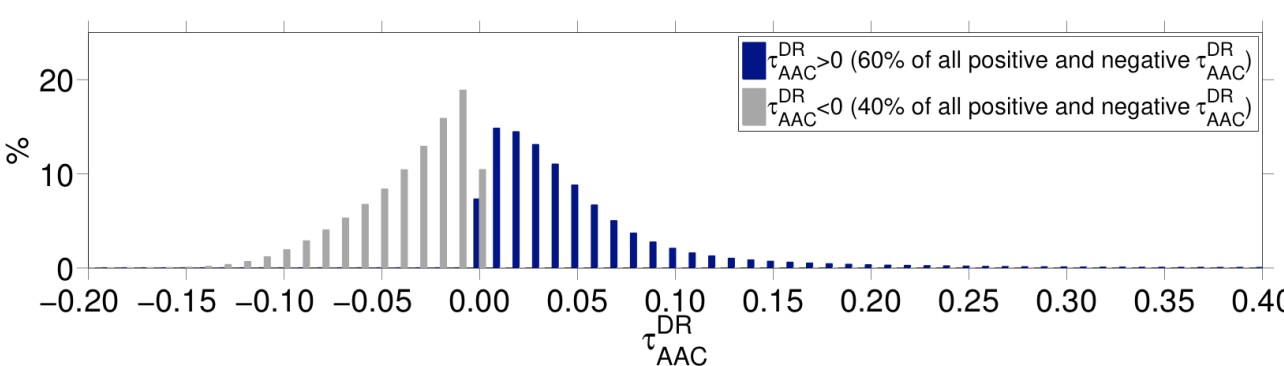

**Figure 3:** Global distribution of $\tau^{DR}_{AAC}$ at 532 nm. Positive (i.e., valid) $\tau^{DR}_{AAC}$ values are in dark blue
(N~3.4M) and negative $\tau^{DR}_{AAC}$ values in grey (N~2.2M). These are nighttime CALIOP measurements
from 2008-2012.
About 40% (i.e. 2.2M data points) of the initial dataset (i.e. N~5.6M) shows negative $\tau^{DR}_{AAC}$ values and
were flagged as invalid data (see Figure 3, in grey). When looking at all valid (i.e. positive) $\tau^{DR}_{AAC}$
values (blue), we show a majority of very small $\tau^{DR}_{AAC}$ values in the 0.01-0.02 AOD range. This agrees
with the findings of Devasthale and Thomas [2011]. Let us note that averaging all data points per 4ºx5º
grid cell (instead of the native resolution shown on Fig. 3) increases the AOD bin of maximum AAC
occurrence globally from 0.01 (Fig. 3) to 0.03.





Table 3 shows four different ways of computing global seasonal and annual averages of aerosol optical
depth above clouds: we use either $\tau^{DR}_{AAC}$ or $\tau^{DR}_{AAC}$ x $f_{AAC}$ (see Case I-II or III-IV) and then either (i)
exclude all cases of $\tau^{DR}_{AAC} < 0$ from the average (i.e., as in Case I and Case III), or (ii) set all cases of
$\tau^{DR}_{AAC} < 0$ to zero, and include these samples in the averages (i.e., as in Case II and Case IV). Let us
note that using $\tau^{DR}_{AAC}$ x $f_{AAC}$ (instead of $\tau^{DR}_{AAC}$) acknowledges the fact that some OWCs present no
overlying aerosols. In this case, we assume that when the DR technique retrieves an invalid AAC
measurement, $f_{AAC} = 0$ and there are no aerosols above the cloud.
**Table 3:** Global seasonal and annual averages of $\tau^{DR}_{AAC}$ (Case I and II) or $\tau^{DR}_{AAC}$ x $f_{AAC}$ (Case III and
IV) when assuming either (i) $\tau^{DR}_{AAC} < 0$ cases are excluded from the averages (Case I and III) or
(ii) $\tau^{DR}_{AAC} < 0$ cases are set to zero and included in the averages (Case II and IV). Annual averages here
(last column) are the mean of the seasonal averages.

| Global mean aerosol optical depth | DJF | MAM | JJA | SON | Annual |
|---|---|---|---|---|---|
| Case I<br>$\tau^{DR}_{AAC}$, invalid $\tau^{DR}_{AAC}$ excluded | 0.04 | 0.05 | 0.05 | 0.05 | 0.05 |
| Case II<br>$\tau^{DR}_{AAC}$, invalid $\tau^{DR}_{AAC}$ =0 | 0.02 | 0.02 | 0.02 | 0.02 | 0.02 |
| Case III<br>$\tau^{DR}_{AAC}$ x $f_{AAC}$, invalid $\tau^{DR}_{AAC}$ excluded | 0.03 | 0.03 | 0.04 | 0.03 | 0.03 |
| Case IV<br>$\tau^{DR}_{AAC}$ x $f_{AAC}$, invalid $\tau^{DR}_{AAC}$ x $f_{AAC}$ =0 | 0.01 | 0.01 | 0.01 | 0.01 | 0.01 |

Figure 4 shows global seasonal nighttime median $\tau^{DR}_{AAC}$ x $f_{AAC}$ from 2008 to 2012 (i.e., as in Case III
of Table 3). The title of each seasonal map (respectively DJF, MAM, JJA, SON) in Figure 4 shows the





global maximum (respectively 0.11, 0.13, 0.22, 0.20), median (0.02 for all seasons) and mean (0.03 in
DJF, MAM and SON and 0.04 in JJA) $\tau^{DR}_{AAC}$ x $f_{AAC}$ values.

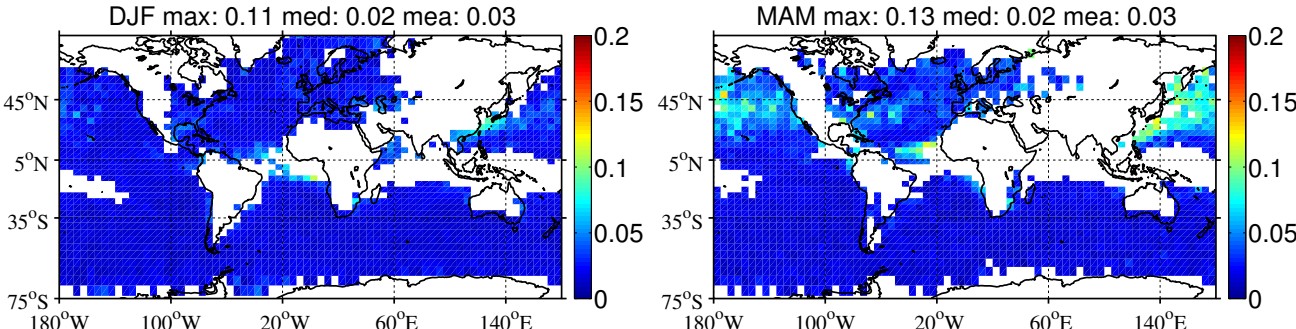

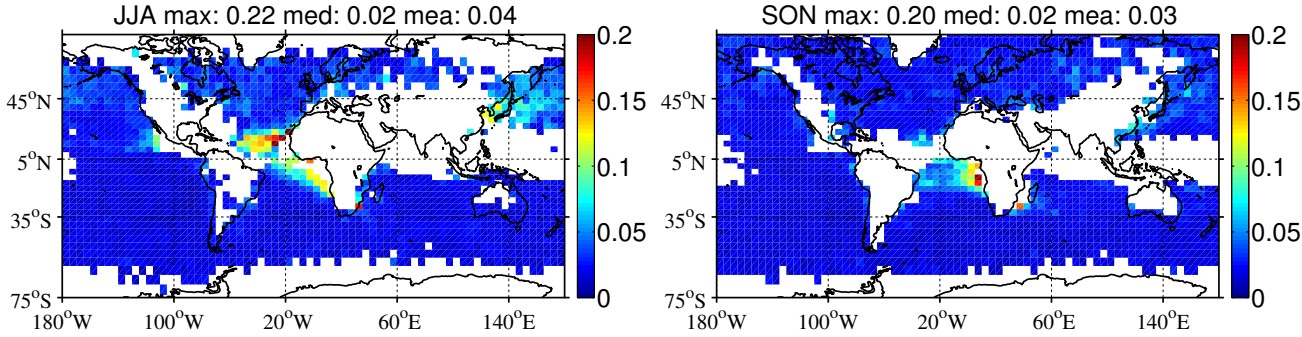

**Figure 4:** Global seasonal 4°x5° nighttime median $\tau^{DR}_{AAC}$ x $f_{AAC}$ from 2008 to 2012. Underlying clouds
satisfy the criteria in Table B2. White pixels show either no CALIOP observations, a limited number of
CALIOP unobstructed OWCs or a limited number of positive $\tau^{DR}_{AAC}$ values. White pixels are not
included when calculating the global mean and median $\tau^{DR}_{AAC}$ values in the title of each map (i.e., as in
Case III in Table 3). Note that if the white pixels were set equal to zero, the seasonal and annual global



$\tau^{DR}_{AAC}$ values would correspond to Case IV in Table 2. The title of each map shows the global
maximum, median and mean values.
We do not expect the $\tau^{DR}_{AAC}$ x $f_{AAC}$ values of Figure 4 to be similar to the results of [Zhang et al., 2014,
Devasthale and Thomas, 2011, Alfaro-Contreras et al., 2016 or Yu and Zhang, 2013] (see Table 2) as
these studies use standard CALIOP Level 2 aerosol and cloud layer products for AAC observations,
instead of using the DR method. On the other hand, the results of Figure 4 seem to be in qualitative
agreement with the global AAC AOD derived from spaceborne POLDER observations [Waquet et al.,
2013a]. Let us note that Waquet et al. [2013a] have to assume an underlying COD larger than 3 to
ensure the saturation of the polarized light scattered by the cloud layer. Although Deaconu et al. [2017]
make different assumptions in the application of the DR method on CALIOP measurements (e.g., they
impose a constant cloud lidar ratio for OWCs with clear air above), they find that POLDER and
CALIOP $\tau^{DR}_{AAC}$ are in good agreement over the SE Atlantic ($R^2 = 0.83$) and over the Tropical Atlantic
($R^2 = 0.82$) from May to October 2008.
**3.2.2. Extinction-to-Backscatter Ratios**
Figure 5 illustrates global seasonal gridded nighttime median AAC extinction-to-backscatter ratio
($S_{AAC}$) values from 2008 to 2012 (section 2.2. describes the calculation of $S_{AAC}$). Bréon [2013] uses
POLDER's specific directional signature close to the backscatter direction to derive aerosol extinction-
to-backscatter values over the globe. Figure 4 of Bréon [2013], although in clear-sky conditions

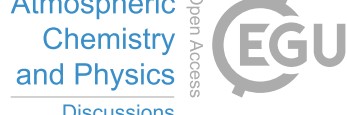

(compared to above OWCs in our case), seems to be in qualitative agreement with Figure 5. However,
Bréon [2013] seems to not detect sufficient aerosol signals in the SE Pacific region to reach any
conclusions.

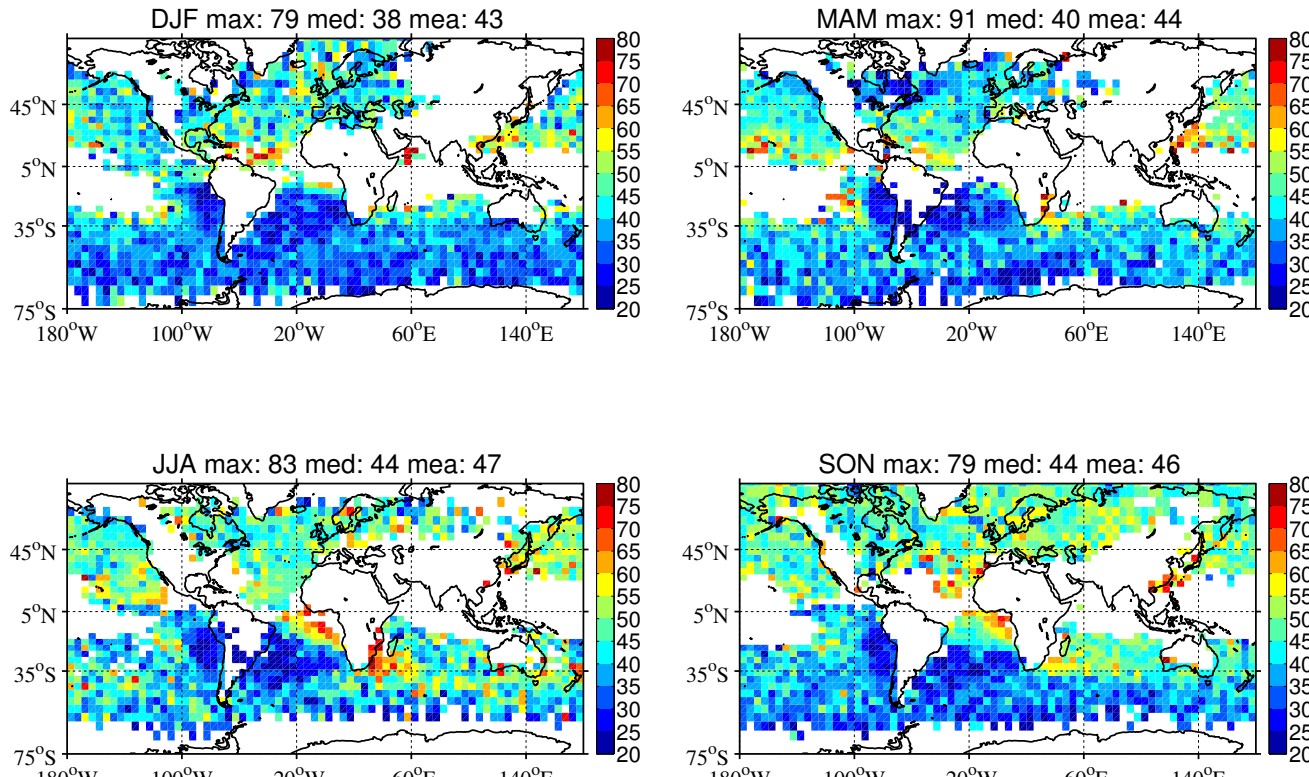

**Figure 5:** Global seasonal 4°x5° nighttime median $S_{AAC}$ at 532 nm (sr) from 2008 to 2012. Underlying
clouds satisfy the criteria in Table B2. White pixels show a limited number of CALIOP OWCs, positive
$\tau^{DR}_{AAC}$ or valid $S_{AAC}$ values (i.e. positive value, the solution has converged and/or the relative difference
in $\tau^{DR}_{AAC}$ is below 0.01). White pixels are not considered in the global mean and median $S_{AAC}$ values in
the title of each map. The title of each map shows the global maximum, median and mean values.



For reference, Table B3 in the appendix lists values of aerosol extinction-to-backscatter (lidar) ratios at 532 nm for different aerosol types (e.g. marine, urban industrial pollution, desert dust, polluted dust, biomass burning) reported in the literature. According to Table B3 and the global mean $S_{AAC}$ values in Fig. 5 (i.e., 43-47 sr in the titles of each map), the aerosol type over OWCs that seems the most common over the globe during nighttime of 2008-2012 is mineral dust. On the one hand, a primary source of aerosols to the TAt region is dust from the Sahara, which can be transported over several thousands of kilometers and reach Central America and the Amazon basin, [Liu et al., 2008, 2015; Herman et al., 1997; Haywood et al., 2003; Waquet et al., 2013a, Zhang et al., 2016]. Over TAt, the season of highest $f_{AAC}$ (i.e., ~80% in Fig. 2) and $f_{AAC} \times \tau^{DR}_{AAC}$ (~0.1-0.2 in Fig. 4) is JJA and this season also shows a mean $S_{AAC}$ of ~50 ± 3 sr (in Fig. 5), which is consistent with the predominance of Saharan dust (see Table B3). On the other hand, a primary aerosol source for the SEAt region is biomass burning from South Africa (see references in Table 1 and 2 for AAC over SEAt). SEAt shows higher mean $S_{AAC}$ values (i.e., above 60 sr in Fig. 5) in JJA, reflecting the presence of biomass burning smoke aerosols (see Table B3). Let us note that $S_{AAC}$ values in our study are slightly lower than in [Liu et al., 2015] (i.e., ~70 sr) over the SEAt region. This is most likely due to our approach to filtering the OWC lidar ratios used in the DR method (see Fig. B3 in the appendix).

**3.2.3 South Atlantic Anomaly Effects**



The South Atlantic Anomaly (SAA) region in Fig. 5, defined within [50ºS, 0ºS; 90ºW, 40ºE], shows
particularly low $S_{AAC}$ results. One would expect to see higher $S_{AAC}$ values in, for example, the SE
Pacific (SEPa) region, as the aerosols in the region are predominantly mixtures of urban/biofuels
(composed of a majority of sulfate aerosols), biomass burning, marine, and/or mixes of smelter
emissions and mineral dust from the Atacama Desert [Chand et al., 2010; Blot et al., 2013]. The SAA is
where the Earth's inner Van Allen radiation belt is the closest to the Earth's surface (at an altitude of
~200 km). This region is characterized by radiation-induced noise spikes in the CALIOP signal that are
especially noticeable at nighttime (Hunt et al., 2009; Noel et al., 2014) and lead to high biases in the
CALIOP integrated attenuated backscatter, which, in turn, lead to low biases in the CALIOP $S_{AAC}$
values in the SAA.
Further investigation has shown (Fig. 6) a lower peak in the $S_{AAC}$ values (~20sr) when these $S_{AAC}$
values are associated with low $\tau^{DR}_{AAC}$ values (i.e., <0.05) and within the SAA region (in purple),
compared to a peak around ~30 sr outside of the SAA region on Fig. 6 (in green).

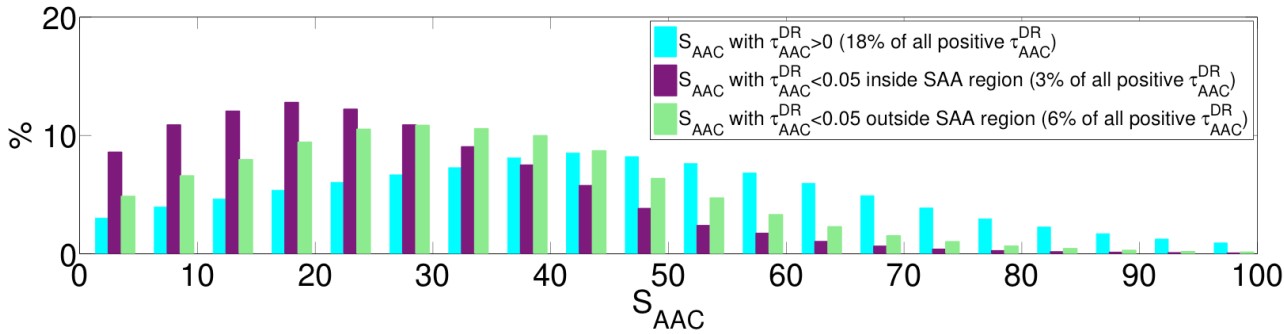



**Figure 6:** Global distribution of $S_{AAC}$ at 532 nm. $S_{AAC}$ values for all positive (i.e., valid) $\tau^{DR}_{AAC}$ values
are in turquoise (N~0.63M, 18% of all positive $\tau^{DR}_{AAC}$ results), $S_{AAC}$ values for $\tau^{DR}_{AAC} < 0.05$ inside the
South Atlantic Anomaly (SAA, defined within [50ºS, 0ºS; 90ºW, 40ºE]) region are in purple (N~0.10M,
3% of all positive $\tau^{DR}_{AAC}$ results) and $S_{AAC}$ values associated to $\tau^{DR}_{AAC} < 0.05$ outside the SAA region
are in green (N~0.22M, 6% of all positive $\tau^{DR}_{AAC}$ results). These are nighttime CALIOP measurements
from 2008-2012.
### 3.3.    AAC Direct Aerosol Radiative Effects
#### 3.3.1.  Global results of DARE$_{OWC}$
Figure 7 shows the seasonal TOA SW DARE$_{OWC}$ estimates (W·m$^{-2}$) that use CALIOP $\tau^{DR}_{AAC}$ x $f_{AAC}$
(see Fig. 4) as input to a radiative transfer model, together with the other parameters described in
section 2.2. DARE$_{OWC}$ in Fig. 7 is set equal to zero (i.e., white pixels) if DARE$_{OWC}$ is invalid or
missing.





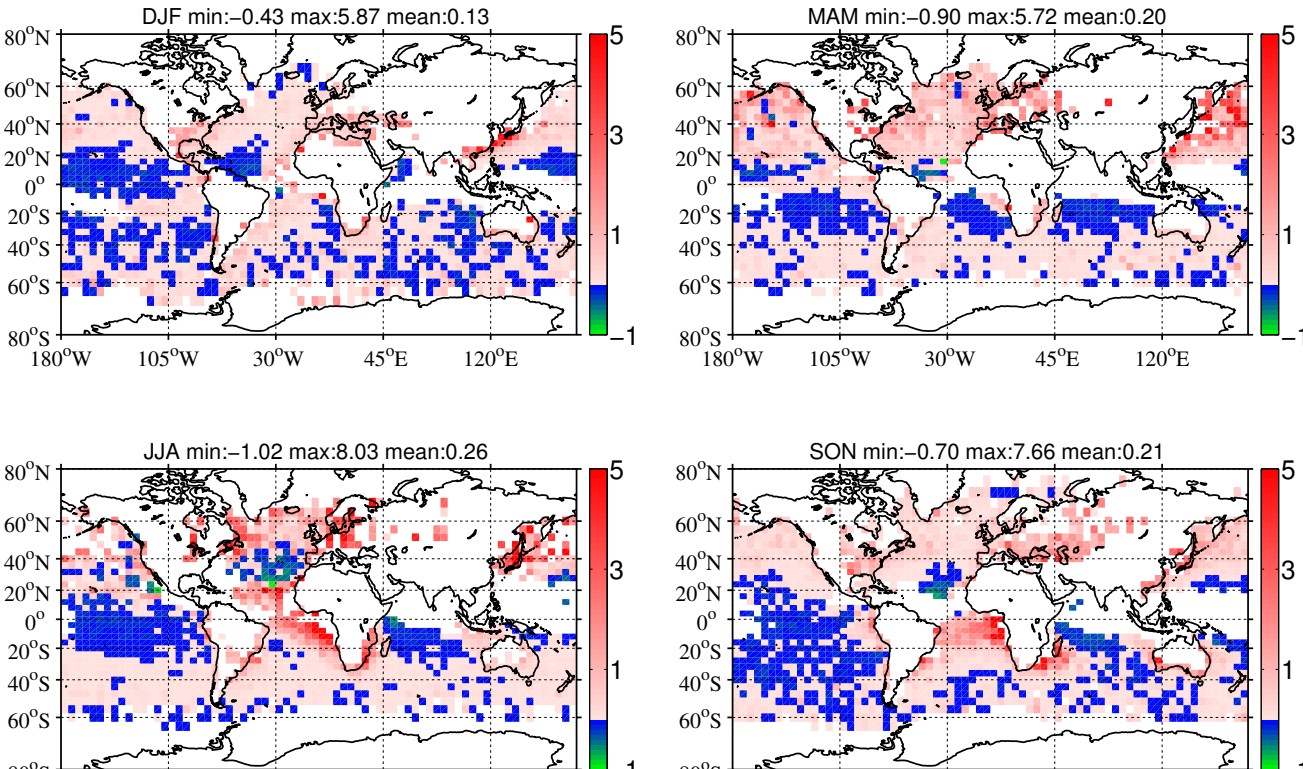

**Figure 7:** Global seasonal 4º×5º TOA SW DARE$_{OWC}$ estimates (W·m$^{-2}$, as described in section 2.2). A white pixel is counted as DARE$_{OWC}$=0 in the global mean DARE$_{OWC}$ values in the title of each map. White pixels show a limited number of CALIOP OWCs, positive $\tau^{DR}_{AAC}$ values or auxiliary MODIS-OMI-CALIOP combined satellite observations. The title of each map shows the global minimum, maximum, and mean values.





Similar to TOA DARE$_{cloudy}$ values from combined A-Train satellites in Oikawa et al. [2013] (see their
Fig. 10) and from General Circulation Models (GCMs) (e.g. SPRINTARS) in Shulz et al. [2006] (see
their Fig. 6 and 7), TOA DARE$_{OWC}$ values in Fig. 7 are mostly positive (i.e., a warming effect due to
less energy leaving the climate system) over the globe. We find, globally, 72% positive 4º×5º
DARE$_{OWC}$ values (i.e., N=4045) against 28% negative values (i.e., N=1581) when considering all four
seasons on Fig. 7. On the other hand, the highest negative TOA DARE$_{OWC}$ values on Fig. 7 (i.e.,
cooling effects shown in green pixels) are over the Tropical Atlantic (in MAM, JJA and SON), in the
Pacific Ocean offshore from Mexico (in JJA) and at the periphery of the Arabian Sea (in JJA).
There are multiple ways to compute the global seasonal and annual DARE$_{cloudy}$ averages (i.e.,
DARE$_{OWC}$ in our case), and it is not clear which method would bring us closer to the true DARE$_{cloudy}$
state of the planet. For this reason, we list several different methods in Table 5. We either use CALIOP
$\tau^{DR}_{AAC}$ or CALIOP $\tau^{DR}_{AAC}$ x $f_{AAC}$ (Case I-II or III-IV) and we either exclude invalid DARE$_{OWC}$ values
or set invalid DARE$_{OWC}$ = 0 (Case I-III or II-IV). For completeness and as an intermediate step towards
DARE$_{all-sky}$ (see Eq. 1), Case V and VI show the global seasonal averages of DARE$_{OWC}$ x Cloud Fraction
(CF), instead of DARE$_{OWC}$. The CF values use monthly MODIS AQUA MYD08_M3 products (variable
"Cloud Retrieva Fraction Liquid FMean"), which are seasonally averaged and 4º×5º-gridded.
**Table 5:** Global seasonal and annual averages of TOA SW DARE$_{OWC}$ estimates (W·m$^{-2}$, as described in
section 2.2). Annual averages (last column) are the mean of the seasonal averages (e.g., 0.53 for Case I
is the average of 0.34, 0.52, 0.71 and 0.56); CF stands for Cloud Fraction.



| Global averaged $DARE_{cloudy}$ $(W \times m^{-2})$ | DJF | MAM | JJA | SON | Annual |
|---|---|---|---|---|---|
| Case I<br>$DARE_{OWC}, \tau^{DR}_{AAC}$, invalid $DARE_{OWC}$ excluded | 0.34 | 0.52 | 0.71 | 0.56 | 0.53 |
| Case II<br>$DARE_{OWC}, \tau^{DR}_{AAC}$, invalid $DARE_{OWC}$=0 | 0.19 | 0.26 | 0.35 | 0.29 | 0.27 |
| Case III<br>$DARE_{OWC}, \tau^{DR}_{AAC}$ x $f_{AAC}$, invalid $DARE_{OWC}$ excluded | 0.24 | 0.40 | 0.53 | 0.40 | 0.39 |
| Case IV<br>$DARE_{OWC}, \tau^{DR}_{AAC}$ x $f_{AAC}$, invalid $DARE_{OWC}$=0 | 0.13 | 0.20 | 0.26 | 0.21 | 0.20 |
| Case V<br>$DARE_{OWC}$ x CF, $\tau^{DR}_{AAC}$, invalid $DARE_{OWC}$ excluded | 0.11 | 0.16 | 0.25 | 0.19 | 0.18 |
| Case VI<br>$DARE_{OWC}$ x CF, $\tau^{DR}_{AAC}$ x $f_{AAC}$, invalid $DARE_{OWC}$=0 | 0.04 | 0.06 | 0.09 | 0.07 | 0.07 |

Global seasonal and annual $DARE_{OWC}$ averages (see titles in Fig. 7 and Table 5) in our study represent
the surface area of each grid cell. Each valid $DARE_{OWC}$ value per pixel on each map of Fig. 7 is
multiplied by the surface of the pixel. These values per grid cell are then summed up and divided by the
sum of the surface of all valid grid cells.
Figure 7 corresponds to the setting of Case IV in Table 5. The reason why we have selected to
showcase this setting is because it closely resembles the settings of the $DARE_{cloudy}$ calculations in
Zhang et al. [2016]; i.e., it assumes DARE = 0 when CALIOP cannot detect an aerosol layer. Figure 7
shows positive global seasonal $DARE_{OWC}$ averages between 0.13 and 0.26 W·m$^{-2}$ (and an annual
average of 0.20 W·m$^{-2}$ in Table 5) as well as the lowest $DARE_{OWC}$ values when compared to $DARE_{OWC}$





values from Case I through Case IV in Table 5. These values are nonetheless much larger than the
global annual ocean $DARE_{cloudy}$ values reported in Zhang et al. [2016] and Schulz et al. [2006] (e.g.,
annual average of 0.015 W $\times$ $m^{-2}$ reported over ocean in Zhang et al. [2016]). Moreover, Matus et al.
[2015] find (see their Table 2) a global TOA $DARE_{cloudy}$ value of 0.1 W·$m^{-2}$ over thick clouds (these
clouds are similar to our study), compensated by a global TOA $DARE_{cloudy}$ value of -2 W·$m^{-2}$ over thin
clouds.
Section 3.3.2 further analyzes $DARE_{OWC}$, together with $f_{AAC}$, $\tau^{DR}_{AAC}$, SSA, and COD results in a few
selected regions and compares these results to previous studies.
### 3.3.2. Regional results of $DARE_{OWC}$
The $f_{AAC}$ results in Fig. 2 help us define six major AAC "hotspots" over the North East Pacific (NEPa),
South East Pacific (SEPa), Tropical Atlantic (TAt), South East Atlantic (SEAt), Indian ocean, offshore
from West Australia (InWA), and North West Pacific (NWPa). To assist in the analysis of the
remaining figures in this study, Figure 8 and Table 6 briefly describe these six AAC hotspots.





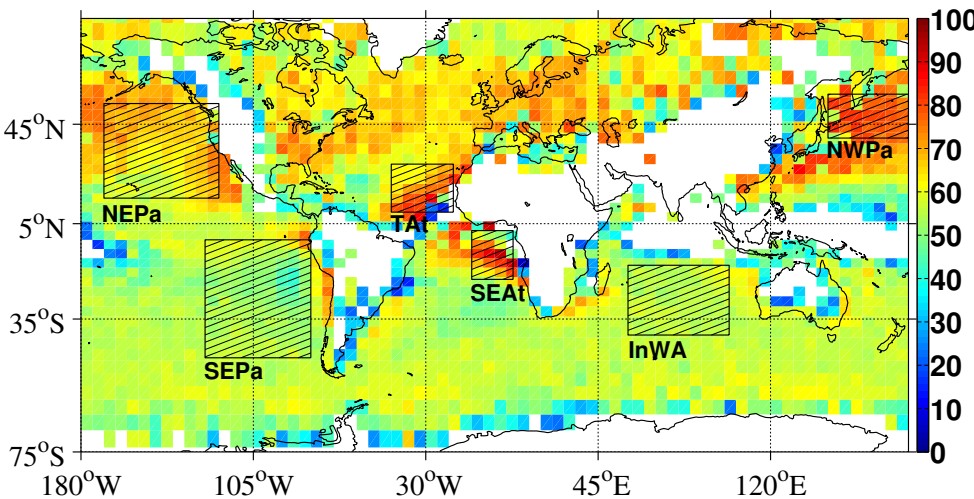

**Figure 8:** Six regions of high AAC occurrence, further defined in Table 6. Background map is the global annual 4º x 5º nighttime AAC occurrence frequency ($f_{AAC}$, see Eq. 3 and Fig. 2 for seasonal $f_{AAC}$ maps). Global annual maximum, median and mean $f_{AAC}$ values are respectively 93%, 57% and 57%.

**Table 6:** Six regions of high AAC occurrence (see Fig. 8), their season of highest AAC occurrence and its corresponding mean $f_{AAC}$ value

| Region | [latitude; longitude] | Season of most $f_{AAC}$ |
|---|---|---|
| North East Pacific Ocean (NEPa) | [16N, 52N; 170W, 120W] | MAM (80%) |
| South East Pacific Ocean (SEPa) | [49S, 2S; 126W, 80W] | DJF (55%) |
| Tropical Atlantic Ocean (TAt) | [10N, 30N; 45W, 18W] | JJA (80%) |
| South East Atlantic Ocean (SEAt) | [19S, 2N; 10W, 8E] | SON (87%) |
| Indian Ocean, offshore from West Australia (InWA) | [41S, 13S; 58E, 102E] | SON (60%) |
| North West Pacific Ocean (NWPa) | [40N, 55N; 145E, 180E] | MAM (90%) |



02

Figure 9a illustrates the mean regional, seasonal or annual estimates of SW TOA $DARE_{OWC}$ ($W \cdot m^{-2}$) in each region of Table 6. Figure 9b-9f show the primary parameters used in the $DARE_{OWC}$ calculations (see section 2.2): the mean regional, seasonal or annual (b) percentage of grid cells that show valid (i.e., positive) $f_{AAC}$ x $\tau^{DR}_{AAC}$ values compared to the total number of 4º x 5º pixels in each region, (c) CALIOP $f_{AAC}$ values, (d) CALIOP $f_{AAC}$ x $\tau^{DR}_{AAC}$ values, (e) assumed overlying SSA values at 546.3 nm and (f) assumed underlying COD values from MODIS.

09



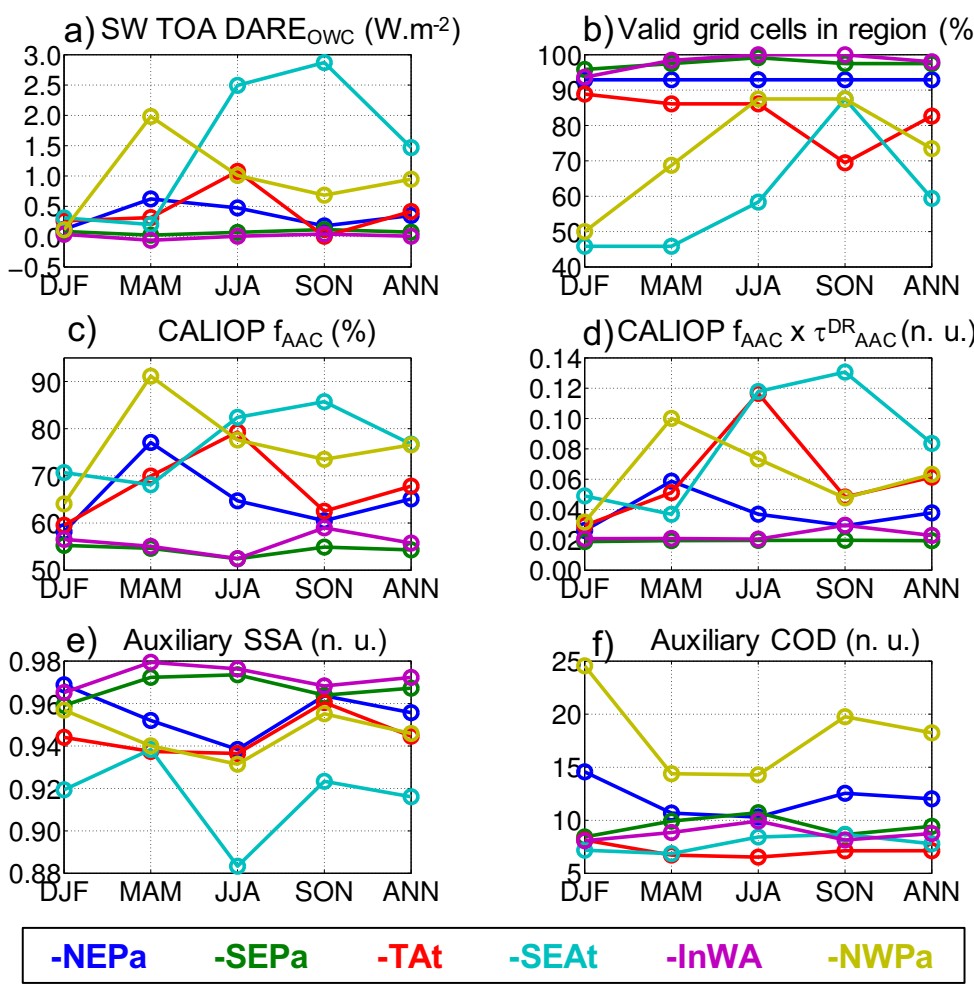

**Figure 9:** Mean regional, seasonal or annual (a) estimated SW TOA $DARE_{OWC}$ (W·m$^{-2}$, calculation is

described in section 2.2), (b) percentage of grid cells that show valid $f_{AAC}$ x $\tau^{DR}_{AAC}$ (i.e., positive)

values compared to the total number of 4° x 5° pixels in each region, (c) CALIOP $f_{AAC}$ (%), (d) $f_{AAC}$ x

$\tau^{DR}_{AAC}$ (no unit), (e) assumed overlying SSA at 546.3 nm from a combination of MODIS-OMI-





CALIOP and (f) assumed underlying COD from MODIS in each region of Table 6. DARE$_{OWC}$ in (a) is
computed using the case IV of Table 5.
Table 7 reports the estimated seasonal or annual, regional range, mean and standard deviations of our
TOA DARE$_{OWC}$ dataset (i.e., values of Fig. 9a)
**Table 7:** Estimated SW TOA DARE$_{OWC}$ (W·m$^{-2}$, setting is case IV of Table 5) in each region of Table

22    6.

| Region | min, max | mean DJF | mean MAM | mean JJA | mean SON | mean ANN |
|--------|----------|----------|----------|----------|----------|----------|
| NEPa | -0.57, 5.10 | 0.12±0.18 | 0.62±0.79 | 0.47±0.78 | 0.18±0.25 | 0.35 ± 0.50 |
| SEPa | -0.21, 2.85 | 0.09±0.19 | 0.02±0.15 | 0.07±0.37 | 0.12±0.44 | 0.07 ± 0.29 |
| TAt | -1.02, 5.25 | 0.26±0.43 | 0.31±0.43 | 1.08±1.66 | 0.01±0.42 | 0.41 ± 0.74 |
| SEAt | 0.20, 7.59 | 0.31±1.09 | 0.20±0.41 | 2.49±2.54 | 2.87±2.33 | 1.47 ± 1.59 |
| InWA | -0.39, 0.83 | 0.04±0.16 | -0.06±0.10 | 0.01±0.11 | 0.04±0.27 | 0.01 ± 0.16 |
| NWPa | 0.07, 5.72 | 0.11±0.14 | 1.98±1.85 | 1.01±1.65 | 0.68±0.46 | 0.95 ± 1.02 |

We record positive TOA DARE$_{OWC}$ values above 1 W·m$^{-2}$ in Fig. 9a over TAt in JJA (1.08 ± 1.66),
SEAt in JJA and SON (2.49 ± 2.54 and 2.87 ± 2.33) and NWPa in MAM (1.98 ± 1.85). Let us note that
the highest positive TOA DARE$_{OWC}$ values on Fig. 9a and in Table 7 may not be entirely representative
of each region, because they are based on a smaller number of valid DARE$_{OWC}$ results (86% valid
values in JJA in TAt, 58-88% in JJA-SON in SEAt and 69% in MAM in NWPa). SEAt and NWPa are
the only regions showing an all-positive range of DARE$_{OWC}$ values in Table 7 (i.e., respectively within



0.20 and 7.59 and within 0.07 and 5.72 W·m$^{-2}$). The spread (i.e., standard deviation) on those mean
regional DARE$_{OWC}$ is of the same order of magnitude as the mean values themselves. For example,
although TAt shows an annual mean DARE$_{OWC}$ value of 0.41 W·m$^{-2}$, most points (i.e., about 68%,
assuming a normal distribution of DARE$_{OWC}$) are within 0.41 ± 0.74 W·m$^{-2}$ (see Table 7). Those regions
and seasons of highly positive DARE$_{OWC}$ values are associated with the highest CALIOP $\tau^{DR}_{AAC}$ x f$_{AAC}$
values (see Fig. 9d: 0.12 in JJA in TAt, 0.12-0.13 in JJA-SON in SEAt and 0.10 in MAM in NWPa).
They are also associated with lower SSA values (i.e., < 0.94 in Fig. 9e), typical of more light absorbing
aerosols such as biomass burning. The underlying COD values are fairly constant (between ~5-10 on
Fig. 9f), except for a noticeably higher COD over the NWPa region (between ~15-25 on Fig. 9f). NWPa
is the region of highest latitudes in our study (i.e., between 40N and 55N). More variation in the COD at
higher latitudes is also observed in Fig. A2 in the Appendix. This agrees with King et al. [2013], who
show a larger zonal variation of COD (and increased uncertainty in the MODIS cloud property
retrievals) in the higher latitudes of both hemispheres, particularly in winter (see their Fig. 12b).
When computing mean DARE$_{OWC}$ results within the "SE Atlantic" region defined in Zhang et al.
[2016] (i.e., [30S, 10N; 20W, 20E] instead of [19S, 2N; 10W, 8E] in our study), we find a small
fraction of valid pixels (i.e., an average of ~37%) but a mean annual DARE$_{OWC}$ value of 0.57 W·m$^{-2}$,
which resides within their range of annual DARE$_{cloudy}$ values (i.e., 0.1 to 0.68 W·m$^{-2}$ in Zhang et al.
[2016]). Similar to Matus et al. [2015], the season of highest DARE$_{OWC}$ is SON over the SE Atlantic
(they find 10% of DARE$_{OWC}$ larger than 10 W·m$^{-2}$ over thick clouds with COD > 1, see their Fig. 9d).
However, our DARE$_{OWC}$ results are significantly higher than the ones in Zhang et al. [2016] in our





SEAt region (defined as a smaller region and offshore from the "SE Atlantic" region in Zhang et al.
[2016]) as well as in the TAt (similar latitude/ longitude boundaries to the ones of region "TNE
Atlantic" in Zhang et al. [2016]) and the NWPa (similar boundaries to "NW Pacific" in Zhang et al.
[2016]) regions.
We emphasize that the $DARE_{OWC}$ estimates in this study are not directly comparable to many previous
studies (see Table 1) because of different spatial domain, period, satellite sensors and associated
uncertainties. This will lead to the detection of different fractions of AAC above different types of
clouds and different AAC types over the globe. The calculations of $DARE_{cloudy}$ can also differ greatly
depending on different AAC aerosol radiative properties assumptions above clouds (especially
absorption) and different assumptions in aerosol and cloud vertical heights (see Table 1).
Apart from the major differences in methods and sensors, it seems reasonable to say that we are missing
AAC cases over pure dust-dominant regions such as the Arabian Sea or the TAt region (compared to
e.g. Zhang et al. [2016] and Matus et al. [2015]). Both Matus et al. [2015] and Zhang et al. [2016] use
the CALIOP Level 2 standard products to distinguish among a few aerosol types and infer specific
aerosol optical properties in their $DARE_{cloudy}$. According to Figure 1(d), SEAt, TAt and the Arabian Sea
are regions where we might be missing up to 40% of AAC cases when using the DR technique
compared to the CALIOP standard products. The number of potentially missing AAC cases in our study
is larger over the Arabian sea ([0-30ºN and 40-80ºE] due to the limited number of OWCs suitable for
the DR method (see section B3 in the Appendix). Zhang et al. [2016] show that pure dust aerosols over
these dust-dominant regions tend to produce a negative $DARE_{cloudy}$ when the underlying COD is below



~7 and this is the case for most of the clouds over these regions in their study. In summary, two factors
in the DR method seem to hamper the detection of AAC in these regions: the low cloud optical depths
of underlying clouds and very few cases of "clear air" above clouds. As a consequence, we propose that
the positive DARE$_{OWC}$ values in our study should, in reality, be counter-balanced by more negative
dust-driven DARE$_{cloudy}$ values over regions such as TAt and the Arabian Sea. On the other hand, the
DARE$_{cloudy}$ results from Matus et al. [2015] and Zhang et al. [2016] might also differ from the true
global DARE$_{cloudy}$ state of the planet for different reasons. As described in Matus et al. [2015], using
CALIOP Level 2 standard products as in Matus et al. [2015] and Zhang et al. [2016] could lead to
possible misclassification of dust aerosols as clouds [Omar et al., 2009], specifically around cloud edges
in the TAt region. Moreover, even if the AAC is correctly detected in Matus et al. [2015] and Zhang et
al. [2016], the amount of AAC AOD might be biased low due to their use of the CALIOP Level 2
standard products [Kacenelenbogen et al., 2014].
**4.   Uncertainties in our DARE above cloud results and the path forward**
**4.1.      Detecting and quantifying the true amount of AAC cases**
Our study uses mainly CALIOP Level 1 measurements to detect aerosols above specific OWCs that
satisfy the criteria given in Table B2. We suggest that the number of CALIOP profiles that contain
aerosols over any type of cloud (instead of only OWCs in this study) should be informed by a
combination of different techniques applied to CALIOP observations (e.g., the standard products, the
DR and the CR technique). Airborne observations such as those from the ObseRvations of Aerosols



above Clouds and their intEractionS (ORACLES) field campaigns are well suited for providing further
guidance on when to apply which technique.
To the best of our knowledge, the true global occurrence of aerosols above any type of cloud remains
unknown. This question cannot be entirely answered with the use of CALIOP observations only. We
suggest that a more complete global quantification and characterization of aerosol above any type of
cloud should be informed by a combination of AAC retrievals from CALIOP, passive satellite sensors
(e.g. POLDER [Waquet et al., 2013a,b, Peers et al., 2015, Deaconu et al., 2017] and MODIS [Meyer et
al., 2013, Zhang et al., 2014, 2016], see Table 2) and model simulations [Schulz et al., 2006].
**4.2.      Considering the diurnal variability of aerosol and cloud properties**
While we consider the diurnal cycle of solar zenith angles in our DARE$_{cloudy}$ calculations, we use
MODIS for underlying COD and cloud R$_e$ information as well as a combination of MODIS, OMI and
CALIOP for overlying aerosol properties (see section 2.2). By using A-Train satellite observations (i.e.,
the AQUA, AURA and CALIPSO platforms), with an overpass time of 1:30 PM local time at the
Equator, we are only using a daily snapshot of cloud and aerosol properties and not considering their
daily variability.
Min and Zhang [2014] show a strong diurnal cycle of cloud fraction over the SEAt region (i.e., a 5-year
mean trend of diurnal cloud fraction using SEVIRI that varies from ~60% in the late afternoon to 80%
in the early morning on their Fig. 4). According to Min and Zhang [2014] (see their Table 2), assuming
a constant cloud fraction derived from MODIS/ AQUA generally leads to an underestimation (less
positive) by ~16% in the DARE$_{all-sky}$ calculations (see Eq. 1). Further studies should explore the



implications of diurnal variations of COD and cloud $R_e$ on $DARE_{cloudy}$ results using, for example,
geostationary observations from SEVIRI.
Daily variations of aerosol (intensive and extensive) radiative properties above clouds cannot be
ignored either. Arola et al. [2013] and Kassaniov et al. [2013] both show that even when the AOD
strongly varies during the day, the accurate prediction of 24h-average $DARE_{non-cloudy}$ requires only daily
averaged properties. However, in the case of under-sampled aerosol properties, such as when using A-
Train derived aerosol properties (this study), the error in the 24h-$DARE_{non-cloudy}$ can be as large as 100%
[Kassaniov et al., 2013]. Xu et al. [2016] show that the daily mean TOA $DARE_{non-cloudy}$ is overestimated
by up to 3.9 W·m$^{-2}$ in the summertime in Beijing if they use a constant MODIS/ AQUA AOD value,
compared to accounting for the observed hourly-averaged daily variability. Kassaniov et al. [2013]
propose that using a simple combination of MODIS TERRA and AQUA products would offer a
reasonable assessment of the daily averaged aerosol properties for an improved estimation of 24h-
$DARE_{non-cloudy}$.
**4.3.    Considering the spatial and temporal variability of cloud and aerosol fields**
We have used coarse resolution (i.e., 4º×5º) seasonally gridded aerosol and cloud properties in our
$DARE_{OWC}$ calculations (see section 2.2). As a consequence, sub-grid scale variability (or heterogeneity)
of cloud and aerosol properties has not been considered. This approach is similar to assuming spatially
and temporally homogeneous cloud and aerosol fields in our $DARE_{OWC}$ results.
Marine Boundary Layer (MBL) clouds show significant small-scale horizontal variability [Di Girolamo
et al., 2010; Zhang et al., 2011]. Using mean gridded COD in $DARE_{cloudy}$ calculations, for example, can





lead to significant biases in DARE$_{cloudy}$ calculations, an effect called the "plane-parallel albedo bias"
[e.g., Oreopoulos et al., 2007, Di Girolamo et al., 2010, Zhang et al., 2011, Zhang et al., 2012]. Min and
Zhang [2014] show that using a mean gridded COD significantly overestimates (by ~10% over the
SEAt region) the DARE$_{cloudy}$ results when the cloud has significant sub-grid horizontal heterogeneity.
Furthermore, this overestimation increases with increasing AOD, COD and cloud inhomogeneity.
Future studies should examine the difference between DARE$_{cloudy}$ results calculated with gridded mean
COD and cloud R$_e$ values (this study) and DARE$_{cloudy}$ results calculated with MODIS Level-3 joint
histograms of MODIS COD and cloud R$_e$ (e.g., similar to Min and Zhang [2014]).
Aerosol spatial variation can be significant over relatively short distances of 10 to 100km, depending on
the type of environment [Anderson et al., 2003; Kovacs, 2006; Santese et al., 2007; Shinozuka and
Redemann, 2011; Schutgens et al., 2013]. Shinozuka and Redemann [2011] argue that only a few
environments can be more heterogeneous than the Canadian phase of the ARCTAS (Arctic Research of
the Composition of the Troposphere from Aircraft and Satellites) experiment where the airmass was
subject to fresh local biomass emissions. In this type of environment, they observed a 19% variability of
the AOD over a 20 km length (comparable in scale to a ~0.1°x0.1° area). They also found a 2%
variability in the AOD over the same length in a contrasting homogeneous environment that occurred
after a long-range aerosol transport event.   As a consequence, similar to using a mean gridded
underlying COD and cloud R$_e$, using mean gridded overlying aerosol radiative properties could very
well bias our DARE$_{OWC}$ results.





As a preliminary investigation into the sources and magnitudes of these potential biases, we have used
TOA $DARE_{non\text{-}cloudy}$ (see Eq. 1) estimates derived using well-collocated aerosol properties (hereafter
called "retrieve-then-average" or R-A) from a companion study (Redemann et al. [2018]; see section A
of the appendix) and compared those to $DARE_{non\text{-}cloudy}$ estimates computed using seasonally gridded
mean aerosol properties at seasonally gridded mean vertical heights (hereafter called "average-then-
retrieve" or A-R). Both $DARE_{non\text{-}cloudy}$ results obtained with the two methods are compared over ocean
and at a resolution of 4º×5º.
A majority (i.e., ~58%) of A-R $DARE_{non\text{-}cloudy}$ results are within ±35% of the R-A $DARE_{non\text{-}cloudy}$
results. We find very few (i.e., ~1%) negative R-A $DARE_{non\text{-}cloudy}$ values paired with positive A-R
$DARE_{non\text{-}cloudy}$ values and very few large differences between both methods (i.e., less than 1% of the
differences are above ±10W m$^{-2}$). However, we find a weak agreement between A-R and R-A
$DARE_{non\text{-}cloudy}$ values during each of the seasons (i.e., a correlation coefficient between 0.21 and 0.34).
The A-R $DARE_{non\text{-}cloudy}$ values are generally biased high relative to the R-A calculations, as illustrated
by positive mean and median values of the A-R to R-A differences (respectively 0.64 W m$^{-2}$ and 0.92
W m$^{-2}$; standard deviation of 2.25). When computing the global seasonal mean A-R and R-A $DARE_{non\text{-}}$
$_{cloudy}$ values separately, we find that the global seasonal A-R $DARE_{non\text{-}cloudy}$ values overestimate the
global seasonal R-A $DARE_{non\text{-}cloudy}$ values by 17%, 19%, 21%, and 17% in DJF, MAM, JJA and SON.
Moreover, the seasonal median A-R $DARE_{non\text{-}cloudy}$ values overestimate the seasonal median R-A
$DARE_{non\text{-}cloudy}$ values in all six regions of Table 6 (i.e., median differences between 0.28 W m$^{-2}$ in
NWPa in SON and 3.05 W m$^{-2}$ in SEAt in JJA). The geospatial distributions of these differences in
DARE calculation strategies are illustrated in Figure 10.





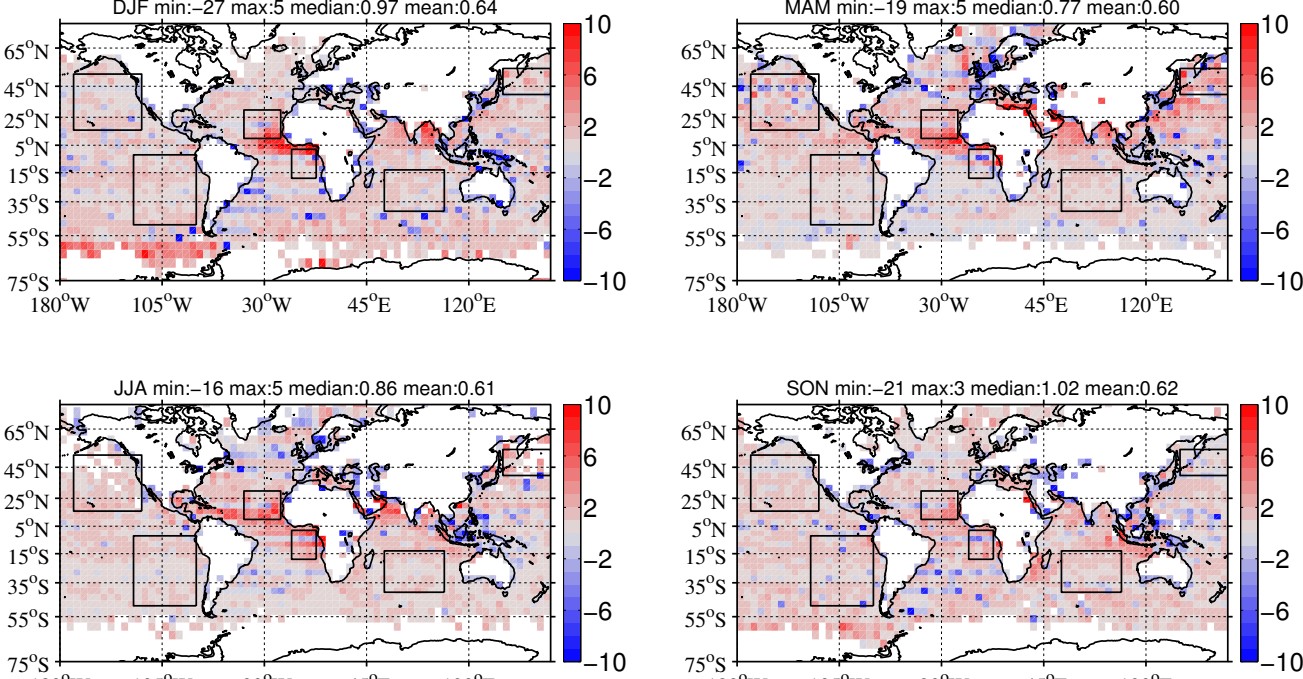

**Figure 10:** Seasonal maps showing the differences in SW TOA DARE$_{non-cloudy}$ computed using the average-then-retrieve (A-R) and the retrieve-then-average (R-A) strategies. Positive values (in red) show regions where the A-R DARE calculations are larger, whereas negative values (in blue) show regions where the R-A DARE calculations are larger. The squares show different regions defined in Table 6. The title of each map shows the global minimum, maximum, median and mean values.

**4.4.        Assuming similar intensive aerosol properties above clouds and in near-by cloud-**
**free skies**





In the calculation of DARE$_{OWC}$, we assume similar intensive aerosol properties above clouds and in
near-by clear skies. This assumption might not be valid and should be investigated in future studies by
comparing aerosol properties and their probability distributions over clear and cloudy conditions using
observations from the ORACLES field campaign.
**4.5.   Assuming fixed aerosol and cloud vertical layers**
Finally, aerosol and cloud layer heights are assumed constant over the globe in our study (see section
2.2). Matus et al. [2015] state that estimates of DARE$_{cloudy}$ over SEAt are highly sensitive to the relative
vertical distribution of cloud and aerosols. Quijano et al. [2000], Penner et al. [2003] and Chung et al.
[2005] demonstrate the importance of the vertical distributions of cloud and aerosol layers in an
accurate estimate of radiative fluxes. Chung et al. [2005], for example, show that varying the relative
vertical distribution of aerosols and clouds leads to a range of global modeled anthropogenic TOA
DARE$_{all\text{-}sky}$ (see Eq. 1) from -0.1 to -0.6 W·m$^{-2}$ (see their Table 2). Future studies should incorporate
mean gridded (i.e., 4°x5° in this study)-seasonal CALIOP Level 2 aerosol and cloud vertical profiles
into the calculation of DARE$_{OWC}$.
**5. Conclusions**
We have computed a first approximation of global seasonal TOA short wave Direct Aerosol Radiative
Effects (DARE) above Opaque Water Clouds (OWCs), DARE$_{OWC,}$ using observation-based aerosol and
cloud radiative properties from a combination of A-Train satellite sensors and a radiative transfer
model. Our DARE$_{OWC}$ calculations make three major departures from previous peer-reviewed results:
(1) they use extensive aerosol properties derived from the Depolarization Ratio, DR, method applied to



Level 1 CALIOP measurements, whereas previous studies often use CALIOP Level 2 standard products which introduce higher uncertainties and known biases; (2) our $DARE_{OWC}$ calculations are applied globally, while most previous studies focus on specific regions of high AAC occurrence such as the SE Atlantic; and (3) our calculations use intensive aerosol properties retrieved from a combination of A-Train satellite sensor measurements (e.g., MODIS, OMI and CALIOP).

Our study agrees with previous findings on the locations and seasons of the maximum occurrence of AAC over the globe. We identify six regions of high AAC occurrence (i.e., AAC hotspots): South and North East Pacific (SEAt and NEPa), Tropical and South East Atlantic (TAt and SEAt), Indian Ocean offshore from West Australia (InWA) and North West Pacific (NWPa). We define $\tau^{DR}_{AAC}$, the Aerosol Optical Depth (AOD) above OWCs using the DR method on CALIOP measurements, $f_{AAC}$, the frequency of occurrence of AAC cases and, $S_{AAC}$, the extinction-to-backscatter (lidar) ratio above OWCs. We record a majority of $\tau^{DR}_{AAC}$ x $f_{AAC}$ values at 532nm in the 0.01-0.02 range and that can exceed 0.2 over a few AAC hotspots. The majority of the $S_{AAC}$ values lie in the $40 - 50$ sr range, which is typical of dust aerosols. $S_{AAC}$ is also consistent with typical dominant aerosol types over the TAt and SEAt regions (respectively dust and biomass burning).

We find positive averages of global seasonal $DARE_{OWC}$ between 0.13 and 0.26 W·m$^{-2}$ and an annual global mean $DARE_{OWC}$ value of 0.20 W·m$^{-2}$ (i.e., a warming effect on climate). Regional seasonal $DARE_{OWC}$ values range from -0.06 W·m$^{-2}$ in the Indian Ocean, offshore from western Australia (in March-April-May) to 2.87 W·m$^{-2}$ in the South East Atlantic (in September-October-November). High





positive values are usually paired with high aerosol optical depths (>0.1) and low single scattering
albedos (<0.94), representative of e.g. biomass burning aerosols.
Although the $DARE_{OWC}$ estimates in this study are not directly comparable to previous studies because
of different spatial domain, period, satellite sensors, detection methods, and/ or associated uncertainties,
we emphasize that they are notably higher than the ones from [Zhang et al., 2016; Matus et al., 2015
and Oikawa et al., 2013].  In addition to differences in satellite sensors, AAC detection methods, and
the assumptions enforced in the calculation of $DARE_{cloudy}$, there are several other factors that may
contribute to the overall higher $DARE_{OWC}$ values we report in this study. The most likely contributors
are (1) a possible underestimate of the number of dust-dominated AAC cases; (2) our use of the DR
method on CALIOP Level 1 data to quantify the AAC AOD; and, in particular, (3) the technique we
have chosen for aggregating sub-grid aerosol and cloud spatial and temporal variability. We discuss
each of these in turn in the following paragraphs.
Two factors seem to be preventing the DR method from recording enough AAC cases in these regions:
the low cloud optical depths of underlying clouds and very few cases of "clear air" above clouds. The
DR method used in this study is restricted to aerosols above OWCs that satisfy a long list of criteria.
The AAC dataset in this study underestimates (i) the total number of CALIOP 5 km profiles that
contain AAC over all OWCs (i.e., not just suitable to the DR technique), (ii) the total number of
CALIOP 5 km profiles that contain AAC over any type of clouds over the globe and (iii) the true global
occurrence of AAC over any type of clouds. To the best of our knowledge, the true amount of AAC in
(i), (ii) and (iii) remains unknown. A better characterization of the "unobstructed" OWCs in the





application of the DR technique on CALIOP measurements might bring us closer to answering (i). A
combination of CALIOP standard, DR and CR techniques together with airborne observations (e.g.,
from the ORACLES field campaign) might answer (ii). Finally, (iii) cannot be answered with the only
use of CALIOP observations. The results in this study should be combined with aerosol-above-cloud
retrievals from passive satellite sensors (e.g. POLDER [Waquet et al., 2013a,b, Peers et al., 2015,
Deaconu et al., 2017] or MODIS [Meyer et al., 2013, Zhang et al., 2014, 2016]) and model simulations
[Schulz et al., 2006] to obtain a more complete global quantification and characterization of aerosol
above any type of clouds.
Compared to other methods, the DR technique applied to CALIOP measurements retrieves $\tau^{DR}_{AAC}$ with
fewer assumptions and lower uncertainties. Other global $DARE_{cloudy}$ results (e.g., Matus et al. [2015]
and Zhang et al. [2016]) use CALIOP standard products to detect the AAC cases, quantify the AAC
AOD and define the aerosol type (and specify the aerosol intensive properties). These studies rely on
the presence of aerosol in concentrations sufficient to be identified by the CALIOP layer detection
scheme, and on the ability of the CALIOP aerosol subtyping algorithm to correctly identify the aerosol
type and thus select the correct lidar ratio for the AOD retrieval. While several recent studies have
taken various approaches to quantifying the amount of aerosol currently being undetected in the
CALIOP backscatter signals, their general conclusions are unanimous. The CALIOP standard products
underestimate above-cloud aerosol loading and the corresponding AAC AOD (Kacenelenbogen et al.,
2014; Kim et al., 2017; Toth et al., 2018; Watson-Parris et al., 2018), and this in turn leads to
underestimates of both $DARE_{non\text{-}cloudy}$ and $DARE_{cloudy}$ (Thorsen and Fu, 2015; Thorsen et al., 2017).





In this study, we have assumed spatially and temporally homogeneous clouds and aerosols in our
DARE$_{OWC}$ calculations. As a preliminary investigation of such effects on our calculations, we have
compared DARE calculations derived from well collocated aerosol properties (retrieve-then-average) to
DARE calculations using seasonally gridded mean aerosol properties (average-then-retrieve). We have
shown that the average-then-compute DARE results generally overestimate the retrieve-then-average
results both on a global scale and in each of our selected regions. Further research and analysis are
required to determine which of these two computational approaches provides the most accurate
estimates                          of                          real-world                          DARE.



**Appendix A: Method to obtain aerosol radiative properties in non-cloudy (i.e., clear-sky)**
**conditions using MODIS, OMI and CALIOP and to estimate DARE$_{non\text{-}cloudy}$**
A companion paper, Redemann et al. [2018], develops and refines a method for retrieving full spectral
(i.e., at 30 different wavelengths) extinction coefficients, Single Scattering Albedo (SSA) and
asymmetry parameters from satellite aerosol products in non-cloudy (i.e., clear-sky) conditions. The
method requires colocation of quality-screened satellite data, selection of aerosol models that reproduce
the satellite observations within stated uncertainties, and forward calculation of aerosol radiative
properties based on the selected aerosol models. They use MODIS-Aqua AOD at 550 and 1240 nm,
CALIPSO integrated backscattering (IBS) at 532 nm and OMI Absorption Aerosol Optical Depth
(AAOD) at 388 nm (see Table A1). The aerosol radiative properties resulting from this method are
called MOC retrievals (for MODIS-OMI-CALIOP).



Table A1. Data sets currently used for global MODIS-OMI-CALIOP (MOC) retrievals of aerosol

radiative properties [Redemann et al., 2018]; DT: Dark Target and EDB: Enhanced Deep Blue.

| Product | Source | Assumed Uncertainties* | Weight*,** |
|---|---|---|---|
| 550 nm AOD | MODIS Collection 6 (Ocean, DT-Land, EDB-Land) | $\pm5\% \pm 30$ Mm$^{-1}$ | 0.1488 |
| 1240 nm AOD | MODIS Collection 6 (extrapolated spectrally over land) | $\pm5\% \pm 30$ Mm$^{-1}$ | 0.1422 |
| 388 nm AAOD | OMI (OMAERO for ocean, OMAERUV for DT-land), MODIS EDB | $\pm30\% \pm 50$ Mm$^{-1}$ | 0.5542 |
| 532 nm IBS | CALIPSO V3-01 | $\pm30\% \pm 0.1$ Mm$^{-1}$sr$^{-1}$ | 0.1548 |

\*    For the values after division by CALIPSO layer depth

\*\* The weight, $w_i$, is used to calculate the cost function $X = (\Sigma w_i((x_i - \hat{x}_i)/\delta\hat{x}_i)^2)^{1/2}$ where $x_i$ are the retrieved parameters,

$\hat{x}_i$ are the observables, $\delta\hat{x}_i$ are the uncertainties in the observables.

The choice of OMI satellite algorithms (see Table A1) reflects their assessment of the

representativeness of subsampling OMI data along the CALIPSO track; i.e., they compared the

probability distribution (PDF) of the OMI retrievals along the CALIPSO track to the global PDF and

chose the data set that had the best match between global and along-track PDF for the over-ocean and

two over-land data sets, the latter being different in their use of MODIS dark target (DT) versus

enhanced Deep-Blue (EDB) data as the source of AOD. They collocate the MODIS and OMI products

within a 40x40 km$^2$ box centered at each CALIPSO 5-km profile location after Redemann et al. [2012].



For the OMAERUV data set, they choose the SSA product for the layer height indicated by the collocated CALIOP backscatter profile.

Their aerosol models emulate those of the MODIS aerosol over-ocean algorithm [Remer et al., 2005]. Like the MODIS algorithm, they define each model with a lognormal size distribution and wavelength-dependent refractive index. They then combine two of these models, weighted by their number concentration, and compute optical properties for the bi-modal lognormal size distribution. Unlike the MODIS algorithm, they allow combinations of two fine-mode or two coarse-mode models. They use ten different aerosol models, which stem from some of the MODIS over-ocean models [Remer et al., 2005] but include more absorbing models, which was motivated by application of their methodology to the Arctic Research of the Composition of the Troposphere from Aircraft and Satellites (ARCTAS) field campaign data, requiring more aerosol absorption than included in the current MODIS over-ocean aerosol models. They use MOC spectral aerosol radiative properties to then calculate Direct Aerosol Radiative Effects (i.e., DARE$_{non\text{-}cloudy}$, see Eq. 1) through a delta-four stream radiative transfer model with fifteen spectral bands from 0.175 to 4.0 μm in SW and twelve longwave (LW) spectral bands between 2850 and 0 cm$^{-1}$ [Fu and Liou, 1992].

In order to use these MOC parameters (retrieved in clear-skies) in our DARE$_{OWC}$ calculations, we need to assume similar aerosol intensive properties in clear skies compared to above clouds and we need to spatially and/ or temporally grid these MOC parameters. As discussed in section 2.2, we use seasonally averaged MOC spectral SSA, aerosol asymmetry parameter, and extinction retrievals on 4°x5° grids. Figure A1 illustrates seasonal maps of MOC SSA used in our calculations of DARE$_{OWC}$.

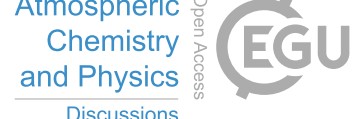



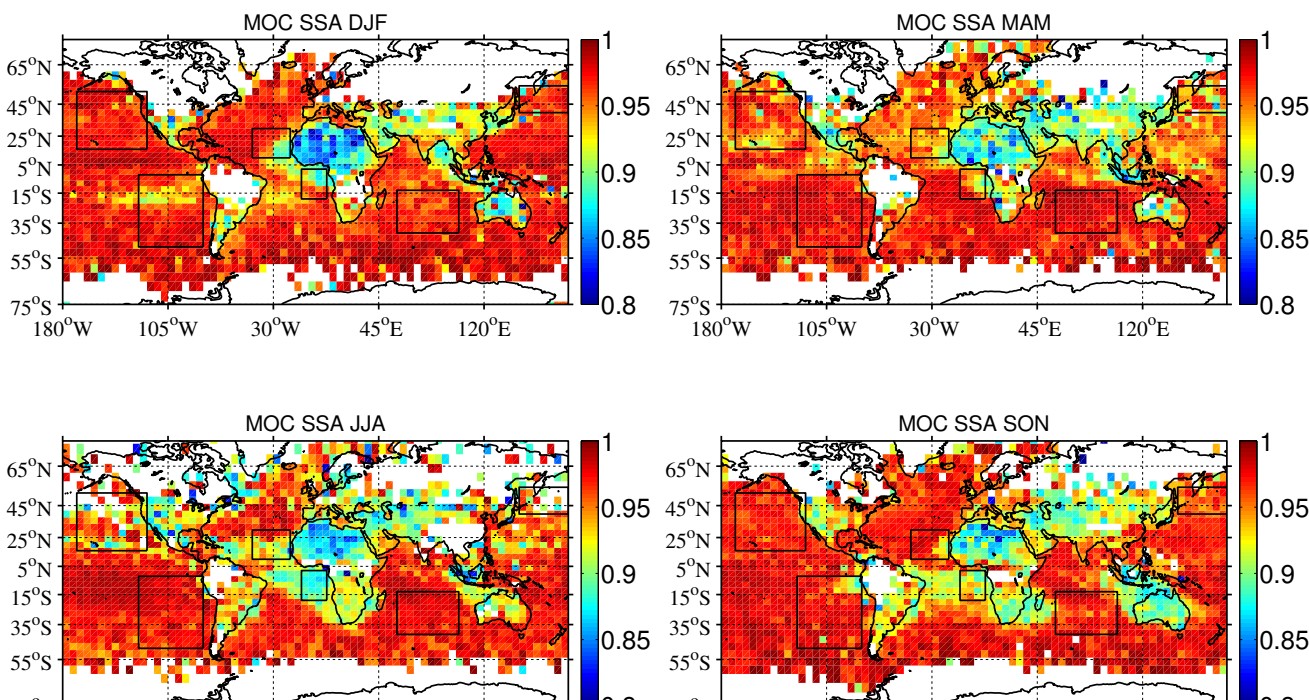

Figure A1: Seasonal maps of MOC SSA at 546.3 nm in 2007 used in the calculations of DARE$_{OWC}$.

The squares show different regions defined in Table 6.

The DARE$_{OWC}$ calculations in our study also require information about the underlying cloud optical

properties. As discussed in section 2.2, we use seasonally mean gridded COD from MODIS such as

illustrated in Figure A2.



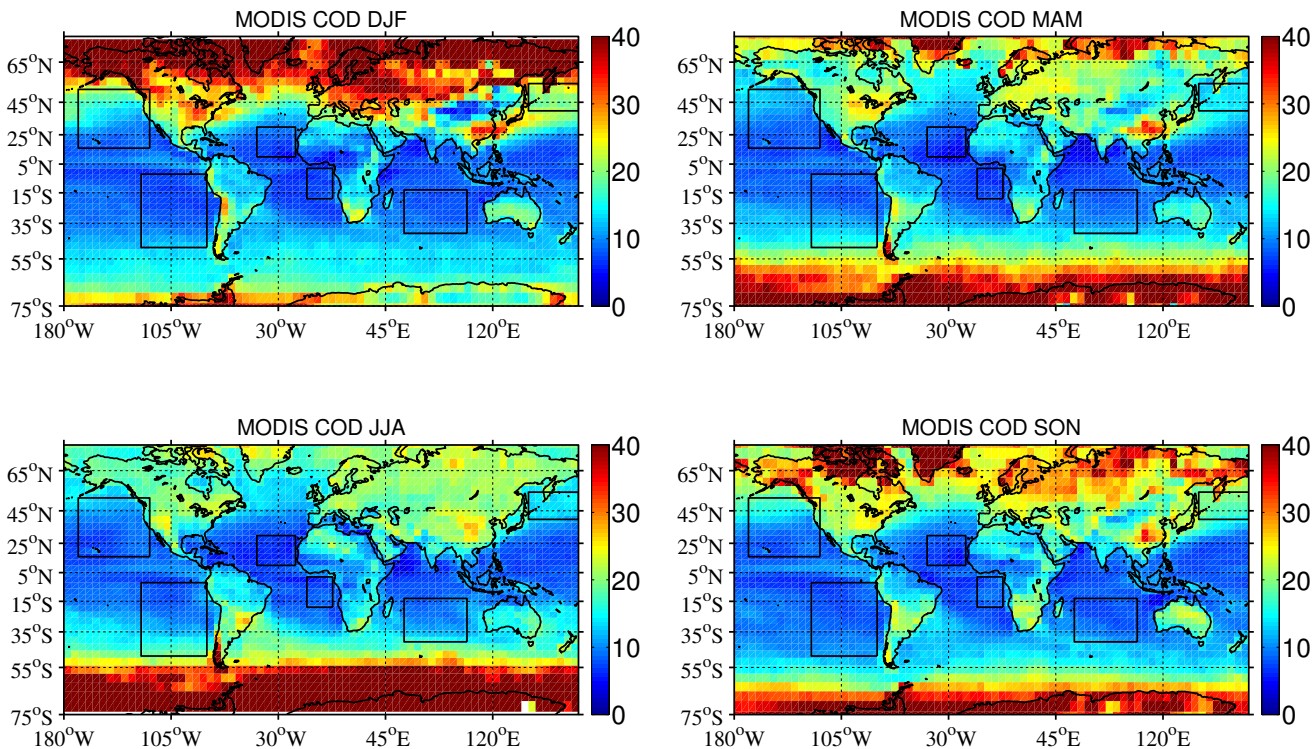

Figure A2: Seasonal maps of COD used in the calculations of DARE_OWC. COD information is inferred

from MODIS seasonally averaged monthly 1°x1° grids (i.e. liquid water cloud products of MYD08_M3:

"Cloud Effective Radius Liquid Mean Mean" and "Cloud Optical Thickness Liquid Mean Mean"

[Platnick et al. 2015]) from 2008 to 2012. The squares show different regions defined in Table 6.

**Appendix B: Method for AAC detection, AAC AOD and $S_{AAC}$ computation**

The depolarization ratio (DR) method [Hu et al., 2007b] used to derive estimates of the optical depths

($\tau$) of aerosols above clouds (AAC) is given in Eq. (2) and repeated here for convenience:

$$\tau^{DR}_{AAC} = -0.5 \times \ln[IAB^{OWC}_{SS,AAC} / IAB^{OWC}_{SS,CAC}] \tag{B1}$$





The subscripts SS and CAC represent, respectively, 'single scattering' and 'clear above clouds'.
$IAB^{OWC}_{SS}$ (i.e., either $IAB^{OWC}_{SS,AAC}$ or $IAB^{OWC}_{SS,CAC}$) is the single scattering integrated attenuated
backscatter (IAB), derived from the product of the measured 532 nm attenuated backscatter coefficients
integrated from cloud top to cloud base, $IAB^{OWC}$, and a layer effective multiple scattering factor, $\eta^{OWC}$,
derived from the layer-integrated volume depolarization ratio of the water cloud (called $\delta^{OWC}$) using:
$\eta^{OWC} = [(1-\delta^{OWC})/(1+\delta^{OWC})]^2$                                                  (B2)
[Hu et al., 2007a]. The single scattering IAB is thus derived using:
$IAB^{OWC}_{SS,X} = \eta^{OWC} \times IAB^{OWC}_{measured,X}$                                   (B3)
for both aerosol above cloud cases (X = AAC) and those cases with clear skies above (X = CAC). An
assumption of the DR method is that $\delta^{OWC}$ is negligibly affected by any aerosols that lie in the optical
path between the OWC and the lidar.
Table B1 provides a high-level overview of the procedure we use to compute aerosol optical depth
($\tau^{DR}_{AAC}$) and aerosol extinction-to-backscatter ratio ($S_{AAC}$) above OWCs over the globe. We chose to
concentrate on night-time CALIOP observations only, as they have substantially higher signal-to-noise
ratios (SNR) than the daytime measurements [Hunt et al., 2009].
**Table B1:** Steps required to compute $\tau^{DR}_{AAC}$ and $S_{AAC}$. (*): we construct global maps of 4 x 5° pixels
using median values. Superscripts 1 and 2 denote respectively CALIOP Level 1 and Level 2 aerosol or
cloud layer products.



| Step | Description | CALIOP, GEOS-5 and other computed products that are used in each step | More detail |
|------|-------------|-----------------------------------------------------------------------|-------------|
| S1 | Select specific Opaque Water Clouds (OWC) suitable for the DR technique | CAD Score[2], Integrated Attenuated Backscatter Uncertainty 532[2], Integrated Volume Depolarization Ratio Uncertainty[2], Horizontal Averaging, Opacity Flag[2], Feature Classification Flags[2], Layer Top Altitude[2], Layer Top Temperature[2], Surface Wind Speed[2] | section B1, Table B2 |
| S2 | Select a subset of OWCs from (S1) with clear air above | Overlying Integrated Attenuated Backscatter 532[2], simulated molecular layer-integrated attenuated backscatter [Powell et al., 2002 and 2006] and OWCs from (S1) | section B2 |
| S3 | Process seasonal maps of median $IAB^{OWC}_{SS,CAC}$ and record number of $IAB^{OWC}_{SS,CAC}$ values per grid cell [(*)] | Integrated Attenuated Backscatter 532[2], Integrated Volume Depolarization Ratio[2], and OWCs with clear air above from (S2) | section B3 |
| S4 | Compute $\tau^{DR}_{AAC}$ along track | Total Attenuated Backscatter 532[1], Molecular Number Density[1], Ozone Number Density[1] Integrated Attenuated Backscatter 532[2,+], Integrated Volume Depolarization Ratio[2,+], Layer Top Altitude[2,+], Layer Base Altitude[2,+] and seasonal maps of $IAB^{OWC}_{SS,CAC}$ from (S3) Note: [(+)] these parameters are re-computed from CALIOP level 1 data, and may differ from the standard CALIOP products | Eq. (2) or Eq. (B1) |
| S5 | Compute $S_{AAC}$ along track | $\tau^{DR}_{AAC}$ from (S4), Total Attenuated Backscatter 532[1] and Molecular Number Density[1] Note: aerosol layer top is set at 12km and aerosol layer base is fixed at the range bin above the recalculated OWC layer top height | Eq. (15), [Fernald et al., 1972] |
| S6 | Process seasonal maps of median $\tau^{DR}_{AAC}$ and $S_{AAC}$ and record number of $\tau^{DR}_{AAC}$ and $S_{AAC}$ values per grid cell [(*)] | $\tau^{DR}_{AAC}$ of (S4), $S_{AAC}$ from (S5) and we filter using number of $IAB^{OWC}_{SS,CAC}$ values per grid cell and per season from (S3) | section 3.2 |



The first step (S1) is to identify OWCs that are suitable for the application of the DR method. The
acceptance criteria used to identify these clouds are described below in section B1 and listed in Table
B2. In the second step (S2), we use the overlying integrated attenuated backscatter (i.e., the 532 nm
attenuated backscatter coefficients integrated from TOA to the OWC cloud tops) to partition the OWC
into two classes: (i) "unobstructed" clouds, for which the magnitude of the overlying IAB suggests that
only aerosol-free clear skies lie above; and (ii) "obstructed" clouds for which we expect to be able to
retrieve positive estimates of $\tau^{DR}_{AAC}$. Section B2 describes the objective method we have developed to
separate unobstructed clouds (for which we can compute $IAB^{OWC}_{SS,CAC}$) from obstructed clouds (for
which we calculate $IAB^{OWC}_{SS,AAC}$).
In step (S3), we construct global seasonal maps of median $IAB^{OWC}_{SS,CAC}$ using 5 consecutive years
(2008-2012) of CALIOP nighttime data (see section B3). By doing this we can subsequently compute
estimates of $\tau^{DR}_{AAC}$ without invoking assumptions about the lidar ratios of water clouds in clear skies
[Hu et al., 2007]. Throughout this study, we chose to compute global median values within each grid
cell (instead of mean values) to limit the impact of particularly high or low outliers on our statistics.
In step (S4), we compute estimates of $\tau^{DR}_{AAC}$ for all obstructed OWC within each grid cell using Eq. (2)
or Eq. (B1) and the 5-year nighttime seasonal median values of $IAB^{OWC}_{SS,CAC}$ from (S3) (i.e., each
$\tau^{DR}_{AAC}$ value along the CALIOP track is computed using one median value of $IAB^{OWC}_{SS,CAC}$ per 4°x5°
pixel and per season).
For the OWCs considered in this study, true layer base cannot be measured by CALIOP, simply
because the signal becomes totally attenuated at some point below the layer top. Instead, what is



reported in the CALIOP data products is an apparent base, which indicates the point at which the signal
was essentially indistinguishable from background levels. Numerous validation studies have established
the accuracy of the CALIOP cloud layer detection scheme (e.g., McGill et al., 2007; Kim et al., 2011;
Thorsen et al., 2011; Yorks et al., 2011; Candlish et al., 2013). Strong attenuation of the signal by
optically thick aerosols above an OWC can, in some cases, introduce biases into the cloud height
determination, which would lead to misestimates of $IAB^{OWC}_{SS,AAC}$ and subsequent errors in $\tau^{DR}_{AAC}$. To
ensure the use of consistent data processing assumptions throughout our retrievals of $\tau^{DR}_{AAC}$ and $S_{AAC}$,
we recalculated the components of $IAB^{OWC}_{SS,AAC}$ (i.e., the "Integrated Attenuated Backscatter 532" and
"Integrated Volume Depolarization Ratio") using parameters in the CALIOP Level 1 product ("Total
Attenuated Backscatter 532", "Molecular Number Density" and "Ozone Number_Density") and
optimized estimates of cloud top and base altitudes based on the "Layer Top Altitude" and "Layer Base
Altitude" values reported in the CALIOP Level 2 layer product.
In step (S5), we compute the $S_{AAC}$ above OWC by solving the two-component lidar equation given by
Eq. (15) of Fernald et al. [1972], and (following Young et al., 2018) reproduced below as Eq. (B4):
$$S_{AAC} = \frac{1 - T^2_{AAC}(0, r_{top}) T_m^{2\frac{S_{AAC}}{S_m}}(0, r_{top})}{2 \int_0^{r_{top}} \beta'(r) T_m^{2\left(\frac{S_{AAC}}{S_m}-1\right)}(0, r) dr} \qquad (B4)$$
$T^2_{AAC}(0,r)$ is the two-way aerosol two-way transmittance between the lidar (at range = 0) and range r. In
our application, $r_{top}$ is the range bin immediately above the OWC top altitude, so that





$T^2_{AAC}(0,r_{top})=\exp(-2x\tau^{DR}_{AAC})$. $T_m(0,r)$ is the one-way transmittance due to molecular scattering and
ozone absorption, $S_m$ is the molecular extinction-to-backscatter ratio, $\beta'(r)$ is the attenuated backscatter
coefficient at range r; i.e.,
$\beta'(r)=(\beta'_m(r)+\beta'_{AAC}(r))xT^2_m(0,r)xT^2_{AAC}(0,r)$                   (B5)
[Young and Vaughan, 2009]. Because the regions studied typically have very low aerosol loading,
molecular scattering often contributes most of the signal hence the two-component lidar equation is
required. Moreover, because equation (B4) is transcendental and cannot be solved algebraically,
solutions are obtained using an iterative method. Valid $S_{AAC}$ values must satisfy $\tau^{DR}_{AAC} > 0$ and $S_{AAC} >$
0, and the iteration much converge to a solution for which the relative difference between successive
$\tau^{DR}_{AAC}$ estimates is less than 0.01 (i.e. $|(\tau^{DR}_{AAC} - \tau^{Fernald}_{AAC})/\tau^{DR}_{AAC}| < 0.01$).
Apart from the identification of specific OWCs in step (S1), the primary Level 2 CALIOP parameters
used to calculate $\tau^{DR}_{AAC}$ (S2-S4 in Table B1) are (i) the integrated attenuated backscatter above cloud
top to detect "clear air" cases (i.e. "Overlying Integrated Attenuated Backscatter 532" in step (S2)), (ii)
the layer integrated attenuated backscatter of the OWC with clear air above (i.e. "Integrated Attenuated
Backscatter 532" in step (S3)) and (iii) the cloud multiple scattering factor, derived as a function of the
layer integrated volume depolarization ratio (i.e. the "Integrated Volume Depolarization Ratio" in S3
and S4).
Below, we list the potential sources of errors associated with those three products:
(a) the accuracy of the 532 nm channel calibrations,



(b) the signal-to-noise ratio (SNR) of the backscatter data within the layer,
(c) the estimation of molecular scattering in the integrated attenuated backscatter (section 3.2.9.1 of the
CALIPSO Feature Detection ATBD, http://www-calipso.larc.nasa.gov/resources/pdfs/PC-SCI-
202_Part2_rev1x01.pdf), and
(d) the accuracy of the depolarization calibration (see section 5 in Powell et al., [2009]).
Concerning (a), Rogers et al. [2011] show that the NASA LaRC HSRL and CALIOP Version 3 532 nm
total attenuated backscatter agree on average within ~3%, demonstrating the accuracy of the CALIOP
532 nm calibration algorithms.
Concerning (b), we assume the influence of the SNR returned from the OWC is negligible as the OWCs
are strongly scattering features and our dataset is composed of nighttime data only. However, the
backscatter from tenuous and spatially diffuse aerosol layers with large extinction-to-backscatter ratios
can lie well beneath the CALIOP attenuated backscatter detection threshold.  When such layers lie
above OWCs, the measured overlying integrated attenuated backscatter can fall within one standard
deviation of the expected 'purely molecular' value that is used to identify CAC (or "unobstructed")
OWC in our dataset (S2; see Sect. B2). Within the context of this study, these tenuous and spatially
diffuse aerosol layers can have appreciable AOD, and thus care must be taken to ensure that these sorts
of cases are not misclassified as CAC OWC. Section B3 discusses such cases, possibly found, for
example, over the region of SEAt.
**B1. Select specific Opaque Water Clouds suitable for DR technique**





Successful application of the DR method (Eq. 2 or Eq. B1) requires a very specific type of underlying
cloud (step (S1) in Table B1). Table B2 lists the criteria we have applied to the CALIOP 5 km cloud
layer products for the selection of these specific OWCs over the globe.
**Table B2:** Criteria used to select the Opaque Water Clouds (OWC) for the application of the DR
method to obtain the AAC frequency of occurrence, AAC optical depth, AAC lidar ratio and $DARE_{OWC}$
in this study.

| criteria | metric | interpretation |
|---|---|---|
| C1 | Number of cloud layers = 1 | a single cloud in each column |
| C2 | High CALIOP cloud-aerosol discrimination (CAD) score ($90 \leq CAD \leq 100$) and high SNR (IAB SNR > 159, $\delta^{OWC}$ SNR > 2) | highly confident of cloud classification |
| C3 | Cloud detected at 5 km averaging resolution with CALIOP single shot cloud cleared fraction = 0 | cloud is spatially uniform over a 5 km averaging interval |
| C4 | CALIOP opacity flag = 1; surface wind speed < 9 m/s | cloud is opaque |
| C5 | CALIOP phase classification is high confidence water; $\delta^{OWC} < 0.5$; cloud top altitude < 3 km; cloud top temperature $\geq$ -10° C | highly confident of cloud phase identification (water) |

We ensure that each cloud is the only cloud detected within the vertical column (C1) and is guaranteed
to be of high quality by imposing filters on various CALIOP quality assurance flags (C2). Imposing the
"single shot cloud cleared fraction = 0" in criterion (C3) assures that the clouds are uniformly detected
at single shot resolution throughout the full 5 km (15 shot) horizontal extent. As a result, we will



intentionally miss any broken clouds and any clouds that show a weaker scattering intensity within one
or more laser pulses with the 15 shot average. On the other hand, enforcing the single shot cloud
fraction = 0 criteria simultaneously ensures that all $\tau^{DR}_{AAC}$ values in this study will lie below a certain
threshold: larger values would attenuate the signal to the point that single shot detection of underlying
clouds is no longer likely.  Consequently, some highly attenuating biomass burning events (e.g., with
$\tau^{DR}_{AAC} >2.5$) can be excluded from the cases considered here.
At high surface wind speeds over oceans, the CALIOP V3 layer detection algorithm may fail to detect
surface backscatter signals underneath optically thick but not opaque layers. In such cases, CALIOP's
standard algorithm may misclassify the column as containing an opaque overlying cloud. To avoid such
scenarios, we exclude all the cases with high surface wind conditions (C4). Let us note that this
condition was applied on the entire dataset, disregarding the surface type (i.e. land or ocean), as our
OWC dataset resides mostly over ocean surfaces (see Figure 1b).
Criterion (C5) requires that the OWC be both low enough (cloud top below 3km) and warm enough
(cloud top temperature above -10ºC as in Zelinka et al. [2012]) to ensure that it is composed of liquid
water droplets. After applying all the criteria of Table B2, the median OWC top height of our dataset is
~1.6 km. According to Hu et al. [2009], any feature showing a cloud layer integrated volume
depolarization ratio above 50% should correspond to an ice cloud with randomly oriented particles.
Criterion (C5) assures the deletion of such cases.
The averaged single-layer, high QA, uniform cloud (i.e. C1-C3 in Table B2) has a top altitude of ~8
km, a top temperature around -38º C and mean surface winds of ~6 m s$^{-1}$. Selecting only those clouds





with top temperatures above -10º C removes 30-40% of the observations.  Subsequently filtering out
clouds with top heights above 3 km removes an additional 30% of the observations. Finally, filtering
out clouds with underlying winds above 9 m s⁻¹ deletes another 20% of the observations. Among all
single-layer, high QA, uniform clouds (i.e. C1-C3 in Table B2), we find that ~45-50% are opaque
clouds (C4), and that ~11-12% satisfy all criteria (C1-C5) of Table B2.
**B2. Select a subset of Opaque Water Clouds with clear air above**
To distinguish between OWCs having clear skies above (i.e., unobstructed clouds, see S2 in Table B1)
and those having overlying aerosols, we examine the overlying integrated attenuated backscatter
reported in the CALIOP Level 2 cloud layer products. The total Integrated Attenuated Backscatter
(IAB) value above a cloud (i.e., $IAB^{tot}_{aboveCloud}$) can be written as follows:
$$IAB^{tot}_{aboveCloud} = \int_0^{cloudtop} \left[ \beta_a(r) T_a^2(0,r) T_m^2(0,r) \right] dr + \int_0^{cloudtop} \left[ \beta_m(r) T_m^2(0,r) T_a^2(0,r) \right] dr \qquad (B6)$$
Here $\beta_a(r)$ and $\beta_m(r)$ are, respectively, the aerosol and the molecular backscatter coefficients (km⁻¹ sr⁻¹)
at range r (km), and $T_a^2(0,r)$ and $T_m^2(0,r)$ are the two-way transmittances between the lidar (at range r =
0) and range r due to, respectively, aerosols and molecules.
Figure B1 shows simulated profiles of the integrated attenuated backscatter above any given altitude, z,
($IAB^{mol}_{above\ z}$) for a purely molecular atmosphere for both daytime (solid green curve) and nighttime
conditions (dashed green curve). These data were generated by the CALIPSO lidar simulator [Powell et
al., 2002; Powell, 2005; Powell et al., 2006] using molecular and ozone number density profiles



obtained from the GEOS-5 atmospheric data products distributed by the NASA Goddard Global
Modeling and Assimilation Office (GMAO). The error envelopes at ±1 standard deviation (light blue
curves) and ±1.5 standard deviation (dark blue curves) around the mean represent measurement
uncertainties for CALIPSO profiles averaged to a nominal horizontal distance of 5 km. The mean
$IAB^{mol}_{above\ z}$ profiles represent an average of all data along the CALIPSO orbit track on 17 March 2013
that began at 03:29:28 UTC and extended from 78.8°N, 20.3°E to 77.3°S, 77.0°W. Spot checks of
mean $IAB^{mol}_{above\ z}$ profiles from different seasons show variations of ~10% or less, depending on
latitude, for altitudes of 3 km and below. The largest differences are found poleward of 30°. While the
daytime and nighttime mean values are, as expected, essentially indistinguishable from one another, the
error envelopes differ drastically due to the influence of solar background noise during daylight
measurements. In this study, we use nighttime measurements only.

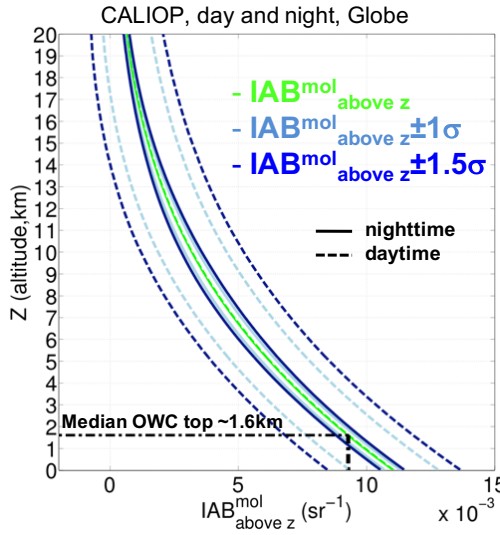

**Figure B1:** Nighttime (solid) and daytime (dashed) simulated vertical profile of integrated attenuated
backscatter above any given altitude, z, $IAB^{mol}_{above\ z}$ (green curve). The light blue (respectively dark



blue) envelope shows 1 (respectively 1.5) standard deviation ($\sigma$) around the $IAB^{mol}_{above\ z}$ profile. Data
was generated by the CALIPSO lidar simulator [Powell et al., 2002 and 2006]. The $IAB^{mol}_{above\ z}$ value
associated to the median OWC top height of ~1.6 km in our dataset corresponds to 0.0093 sr$^{-1}$.
In this study, we assume "clear air" when $IAB^{tot}_{aboveCloud}$ is within the simulated $IAB^{mol}_{aboveCloud}$ value $\pm$
1$\sigma$ (i.e., the light blue envelope shown in Figure B1). This definition of "clear air above" conditions is
somewhat more restrictive than those imposed in previous studies. For example, Liu et al. [2015]
conducted an extensive study of AAC optical depths and lidar ratios using CALIOP measurements over
the tropical and southeast Atlantic. To identify clear air above cloud cases, Liu et al. [2015] require that
the integrated attenuated scattering ratio, defined as

$$ASR = \frac{\int_{8km}^{OWC_{top}} (\beta_m(r) + \beta_a(r)) T_m^2(0,r) T_a^2(0,r) dr}{\int_{8km}^{OWC_{top}} \beta_m(r) T_m^2(0,r) dr} \qquad (B7)$$
, fall within the range of 0.95 < ASR < 1.05, irrespective of cloud top altitude. For comparison, at the
maximum OWC top altitude used in our analyses (3 km), ($IAB^{mol}_{aboveCloud} \pm 1\sigma$) / $IAB^{mol}_{aboveCloud}$ = 1 $\pm$
0.0380. This restriction tightens for lower cloud top heights; e.g., at our mean OWC top altitude of 1.6
km, ($IAB^{mol}_{aboveCloud} \pm 1\sigma$) / $IAB^{mol}_{aboveCloud}$ = 1 $\pm$ 0.0325.
The pioneering study by Chand et al. [2008], who first used the CALIOP DR method to assess the
radiative effects of aerosols above clouds, took a different approach to identifying "clear above cloud"
cases. Rather than examining the overlying IAB, they instead assumed clear air above conditions



whenever $IAB^{OWC}_{SS} > 0.025$ sr$^{-1}$. As will be shown in section B3, in addition to the $IAB^{mol}_{aboveCloud}$
limits cited above, our study also enforces limits on $IAB^{OWC}_{SS,CAC}$. This combination of limits on both
$IAB^{mol}_{aboveCloud}$ and $IAB^{OWC}_{SS,CAC}$ serves to more effectively reject aerosol-contaminated profiles from
the "clear above" data set than either one alone.
**B3. Process median seasonal maps of Integrated Attenuated Backscatter of Opaque Water Clouds**
**showing Clear Air Above**
Once we select specific OWCs (i.e., that satisfy the criteria of Table B2) and define which ones are
"unobstructed" (see section B2), we can easily compute $IAB^{OWC}_{SS,CAC}$ by using Eq. (B3). For clouds
that totally attenuate the lidar signal (i.e., cloud optical depths greater than ~6 [Young et al., 2018]),
$IAB^{OWC}_{SS,CAC}$ in Eq. (2) or Eq. (B1) is related to the OWC lidar ratio (called $S_c$), so that
$S_c = 1 / (2 \times \eta^{OWC} \times IAB^{OWC}_{CAC}) = 1 / (2 \times IAB^{OWC}_{SS,CAC})$          (B8)
[Platt, 1973]. OWC $S_c$ values are relatively stable at the visible and near infrared wavelengths [Pinnick
et al., 1983, O'Connor et al., 2004], but show large variations over land [Pinnick et al., 1983; Hu et al.,
2006]. $S_c$ is known to vary as a function of cloud droplet microphysics, and is especially sensitive to
cloud droplet effective radius ($R_e$) and the imaginary part of the refractive index (see Fig. 8 of Deaconu
et al. [2017]). Hu et al., [2006], Liu et al. [2015] and Deaconu et al. [2017] show that a decrease of $R_e$ is
often paired with an increase of estimated $S_c$ at 532 nm for pure, non-aerosol-contaminated water
clouds (i.e., cloud droplets having an imaginary refractive index of 0).





As an example, Figure B3a shows the median nighttime CALIOP $S_c$ values over the globe during 2008.
Figure B3b shows MODIS AQUA-derived mean liquid water $R_e$ in 2008 (using MODIS Level 3
monthly product "Cloud Effective Radius Liquid Mean Mean").

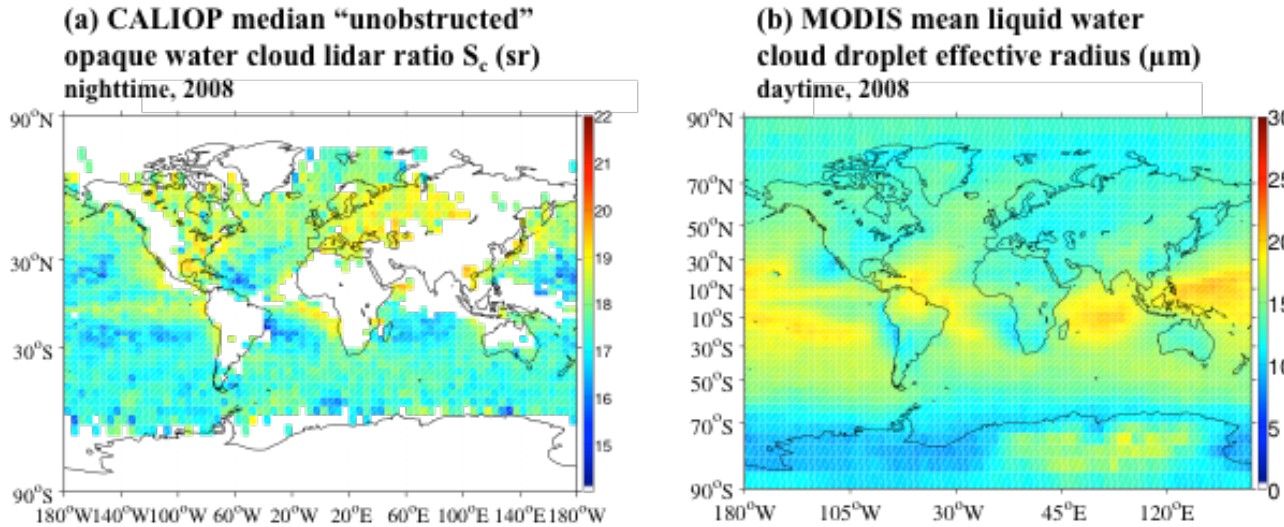

**Figure B3:** a) Global CALIOP yearly median nighttime "unobstructed" (i.e. clear air above) OWC lidar
ratio, $S_c$, in 2008 that satisfy all criteria of Table B2. For the reasons outlined in this section, any OWC
along the CALIOP track for which $S_c > 20$ sr or $S_c < 14$ sr is deleted before temporal and spatial
averaging. White pixels show a limited number of OWCs; b) Global MODIS yearly mean daytime
liquid water cloud droplet effective radius, $R_e$ (in µm, "Cloud Effective Radius Liquid Mean Mean"
parameter from MODIS MYD08_M3 product).



Greater $S_c$ values paired with lower cloud $R_e$ can be seen offshore and close to the west coasts of Africa
and the Americas on Figure B3. Other notable regions of low cloud $R_e$ and high $S_c$ on Figure B3 are
above industrial regions like northern Europe, the eastern US and South East Asia. These results appear
to support Twomey's analysis [Twomey, 1977; Rosenfeld and Lensky, 1998], showing an enhancement
of the cloud albedo through the increase of droplet number concentration and a decrease in the droplet
size driven by increased aerosol concentration. On the other hand, Figure B3a mostly exhibits low $S_c$
values (paired with large $R_e$) over the inter-tropical convergence zone (ITCZ), likely associated with
deep convective regimes. In addition, Figure B3a generally shows larger OWC $S_c$ values in the northern
hemisphere than in the southern hemisphere, which we attribute to differences in sources of cloud
condensation nuclei. Figure B3b shows patterns that are generally similar to those in Figure B3a, but of
opposite intensity. Let us note that the polarization measurements from the space-borne POLDER
sensor [Deschamps et al., 1994] were also used to estimate $R_e$ of liquid water clouds over the globe
[Bréon and Colzy, 2000] and seem to be in qualitative agreement with the findings of Figure B3b.
During our assessment of 5 years of CALIOP data over the globe, we have observed significantly
higher "unobstructed" OWC $S_c$ values (i.e., $S_c > 20$ sr, not shown on Fig B3a) near the coasts of West
Africa and over the region of SE Asia (e.g., see Young et al., [2018]). These may be physically
plausible and either (1) associated with small cloud $R_e$, resulting from the Twomey's effect as explained
above or (2) associated with the presence of light-absorbing aerosols residing within the OWCs
[Mishchenko et al., 2014; Chylek and Hallett, 1992; Wittbom et al., 2014]. These aerosols would be
undetected in our $IAB^{mol}_{aboveCloud}$ clear air selection method (see section B2) and would impact the
chemical composition of the cloud droplets, modifying their backscattered light. The latter is well



illustrated in Fig. 8 of Deaconu et al. [2017], which shows simulations of cloud $S_c$ with an imaginary
part of the refraction index equals to 0.0001, as a function of cloud droplet effective radius. Other
reasons for these unusually high $S_c$ values could be the sources of uncertainty noted (a), (b), (c) and (d)
in the beginning of section B, with (c) (i.e., the SNR of the backscatter data within the layer) possibly
having a much higher impact on $S_c$ than all other factors. An additional source of uncertainty on the
retrieval of $S_c$ could be a failure of the CALIPSO surface detection scheme. If CALIOP fails to detect
the surface adequately, part of the Earth's surface could be misclassified as an opaque water cloud and
these misclassified clouds would have abnormally high $S_c$.
Let us note that the vast majority of the $S_c$ values reported in the literature (i.e., in Hu et al., [2006], Liu
et al. [2015] and Deaconu et al., [2017]) are estimated using a Mie code and not directly measured.
However, none of these results show $S_c$ values above 20 sr for non-aerosol-contaminated OWCs. On the
other hand (and to add a lower bracket on our OWC $S_c$ calculations), none of these results show $S_c$
values below 14 sr. For this reason, we have imposed an additional threshold on the OWC $S_c$ values as
part of step (S3) in Table B1: we delete any "unobstructed" OWC along the CALIOP track for which $S_c$
> 20 sr (i.e., unrealistically small water cloud droplets) or an $S_c$ < 14 sr (i.e., unrealistically large water
cloud droplets). Every OWC $S_c$ value along the CALIOP track was then compiled to produce four
global median seasonal 4º×5º maps of OWC $S_c$ using 5 years of night-time CALIOP data (from 2008 to

81 2012).





There is additional precedent for establishing an upper limit of $S_c$ = 20 sr. Note that, from Eq. B8, the
value of $IAB^{OWC}_{SS,CAC}$ corresponding to $S_c$ = 20 sr is 0.025 sr$^{-1}$. As mentioned earlier, this is the same
OWC IAB threshold value used by Chand et al. [2008] to identify their "clear air above" cases.
**B4. Extinction-to-Backscatter (Lidar) Ratio**
Table B3 lists some typical, recently reported values of the aerosol lidar ratios ($S_a$) measured for various
aerosol types. These data include CALIOP retrievals for several species (e.g., marine, dust, and smoke)
as well as ground-based measurements made using high spectral resolution lidars (HSRL) and Raman
lidars.
**Table B3:** retrieved aerosol extinction-to-backscatter ratios ($S_a$) reported in the literature (PBL:
Planetary Boundary Layer)

| $S_a$ (532 nm, sr) | Aerosol type, $S_a$ value and references (non-exhaustive) |
|---|---|
| 20-25 | **Marine** PBL North Atlantic and PBL tropical Indian ocean 23 sr [Müller et al., 2007] |
| 26-30 | **Marine** global ocean 26 sr [Dawson et al., 2014]; Mix of **Marine** and **Pollution**, case study offshore East Coast USA 26.3 sr [Josset et al., 2011] |
| 31-35 | **Gobi dust** Beijing PBL 35 sr [Müller et al., 2007]; Mix of **Marine and dust**, two case studies Caribbean, 32-33 sr [Josset et al., 2011] |



| 36-40 | **Arabian dust** 33.7 ± 6.7 to 39.1 ± 5.1 sr [Mamouri et al., 2013]<br>**Sahara dust** 39.8 ± 1.4 sr [Omar et al., 2010] |
|---|---|
| 41-45 | **Urban** South Africa 41±13sr [Giannakaki et al., 2016]<br>**Desert dust** Middle East 42.6 sr and India 43.8 sr [Schuster et al., 2012];<br>**Desert dust** Tropical North Atlantic 45.1±8.8 sr [Liu et al., 2015] |
| 46-50 | **Desert dust** African Sahel 49.7sr [Schuster et al., 2012] |
| 51-55 | **Desert dust** PBL 55 sr [Müller et al., 2007];<br>**Urban** Haze central Europe 53 sr [Müller et al., 2007];<br>**Asian dust** 51 sr [Liu et al., 2002] |
| 56-60 | **Desert dust** non-Sahel North Africa 55.4 sr [Schuster et al., 2012];<br>**Desert dust** Africa 60 sr [Pedrós et al., 2010] |
| 66-70 | **Biomass burning** South East Atlantic 70.8±16.2 sr [Liu et al., 2015] |
| 71-85 | **Biomass burning** South Africa 75±14sr [Giannakaki et al., 2016] |



**Data Availability:**

This study used the following A-Train data products: (i) CALIPSO version 3 lidar level 1 profile products (Powell et al. [2013]; NASA Langley Research Center Atmospheric Science Data Center; https://doi.org/10.5067/CALIOP/CALIPSO/CAL_LID_L1-ValStage1-V3-01_L1B-003.01; last access: 26 September 2018), (ii) CALIPSO version 3 lidar level 2 5 km cloud layer products (Powell et al. [2013]; NASA Langley Research Center Atmospheric Science Data Center; https://doi.org/10.5067/CALIOP/CALIPSO/CAL_LID_L2_05kmCLay-Prov-V3-01_L2-003.01; last access: 26 September 2018), (iii) MODIS Atmosphere L2 Version 6 Aerosol Product (Levy and Hsu [2015]; NASA MODIS Adaptive Processing System, Goddard Space Flight Center, USA; http://dx.doi.org/10.5067/MODIS/MOD04_L2.006; last access: 26 September 2018), and (iv) L2 Version 3 OMI products OMAERO [Stein-Zweers and Veefkind, 2012] and OMAERUV [Torres, 2006].

**Author contributions:**

The overarching research goals were formulated by Dr Redemann. Dr. Kacenelenbogen, Dr. Young and Mr. Vaughan influenced the evolution of these research goals. Dr. Kacenelenbogen carried out the formal analyses, investigations and visualizations and wrote the original draft. All co-authors have reviewed and edited the multiple drafts of the manuscript. The methodology behind the global application of the DR method to CALIOP measurements was first developed by Dr. Hu, and adapted by Dr. Kacenelenbogen, Dr. Young, Mr. Vaughan, and Ms. Powell to accommodate the requirements of



this study. The methodology for using this combination of A-Train satellites to infer aerosol intensive
radiative properties was conceptualized by Dr Redemann. The joint MODIS-OMI-CALIOP aerosol
radiative properties were developed and provided by Dr. Shinozuka, Mr. Livingston and Ms. Zhang. Dr.
LeBlanc performed the radiative transfer calculations that provided Direct Aerosol Radiative Effects
estimates in clear skies and above clouds.
**Competing interests:**
The authors declare that they have no conflict of interest.
**Acknowledgements:**
We thank the CALIPSO lidar science working group and data management team for their efforts in
providing and discussing these data sets. We appreciate the comments of the reviewers that have helped
us to improve the paper. We are grateful for comments from Dr. Zuidema and Dr. Wood on cloud
microphysics over the South East Atlantic. This study was funded in part by NASA's Research
Opportunities in Space and Earth Sciences (ROSES) program under grant NNH12ZDA001N-CCST.



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
