# Peer review of "Using a Combination of A-Train Satellite Sensors"

_Atmospheric Chemistry and Physics, 2018_

## Referee Comment (RC1) · Anonymous Referee #3 · 30 Dec 2018

This paper presents a new estimate of the shortwave direct radiative effect attributable to aerosols above clouds. The paper builds on the previous literature by advancing a technique that applies globally and utilizes the depolarization ratio method applied to global CALIOP observations. Use of the depolarization ratio method improves upon a widespread underestimate of aerosol optical thicknesses in the standard CALIOP retrieval products, but seems to struggle to capture cases of dust over clouds. The paper does a nice job of summarizing the studies that have come before, the variety of methods that have been applied to this problem, and the problems that hinder the

precise quantification of the global radiative effect. The paper also does a nice job of placing their quantitative results in the context of other estimates. I have only some minor comments. After addressing these, the paper should be suitable for publication in ACP.

1) The analysis is restricted to clouds that are determined to be opaque, but the method by which opaque clouds are distinguished from clouds that are not opaque is not clear. In the appendix it is noted that the "CALIOP opacity flag" is used. There should be a brief mention in the body of the paper of the physical basis for the "opacity flag". Are there particular regimes where low clouds are prevalent but transparent? There is a vague reference to "clouds such as the ones reported in Leahy et al. (2012)". A more specific description would be better.

2) Figure 1 indicates that the Southern Ocean is the most prominent place on the globe for uniform single layer clouds, but panel b suggests that they are not suitable for the depolarization ratio method. Perhaps it is not of great importance if most of this region has little appreciable aerosol above the cloud layer. Nevertheless, I was left wondering why. Is it a quality of the clouds? Or merely a lack of aerosol optical thickness?

3) Panel d of figure 1 shows a substantial underestimate of cases of aerosol above cloud compared to a similar statistic based on the standard CALIOP aerosol optical thickness product for continents and for oceanic regions dominated by dust plumes. This is discussed in a couple of places in the manuscript, but nevertheless I remained confused as to the cause. The only indication in the body of the paper on line 316 where it says "...filtering out of 'unobstructed' but potentially aerosol-contaminated OWCs." The paper does not make clear what "obstructed" or "unobstructed" means in this context or why such clouds would be filtered. This sentence is in dire need of some plain English.

4) Another place where the description is so technical as to hide the point is in the discussion of the extinction-to-backscatter ratios in sections 3.2.2 and 3.2.3. My sense

is that there is an important point in these sections and that differences in the probability distributions in figure 6 must be significant. But it was not clear what that point is or what the significance to the main result of the paper is.

5) Minor point: In the sentence beginning in line 308 the authors state "...negative (positive) values in blue (red) show the number of AAC cases that are missed (gained)..." Way back in 2010 Prof. Robock pleaded with us to end this misuse of parentheses [Robock, A. (2010), Parentheses are (are not) for references and clarification (saving space), Eos Trans. AGU, 91(45), 419–419, doi:10.1029/2010EO450004]. My understanding is that one of the publishers in our field has specifically written it out of their style guide. I read pretty widely and the only genre of writing where I have experienced this application of parentheses is in the atmospheric sciences journals. I hope the authors will consider rewriting this sentence.

---

## Referee Comment (RC2) · Devasthale (Referee) · 24 Jan 2019

Norrköping, 2019-01-24

I will keep the review short and to the point. If not the lengthiest, it is one of the lengthiest manuscripts I have reviewed so far. So it took me some time to go through it few times and come to the grips of how the DARE_OWCs are actually computed. But once I stated reading it carefully, it was easier to follow and understand. I appreciate the hidden efforts behind the work needed to bring onboard information from the suit of

sensors. I also appreciate the way authors contrast and compare their results with the previous studies. Table 5 is a good idea and could be useful for evaluating models. As far as the methodology and results are concerned, I do not see anything that should raise a red flag. I do however have one key concern as mentioned below.

CALIOP offers two distinct advantages over passive sensors, namely its superiority in detecting aerosol layers and their precise altitudes. While the authors go to such a great length and detail to be as realistic and up-to-date in taking into account aerosol and cloud layers (and their properties) as possible, if I am not mistaken, the altitude of these layers is assumed to be constant globally. And I can't help but wonder how this is going to affect their estimates, given the diversity in the verticality of aerosol and clouds in the AAC scenarios and its impact on DARE_OWCs. It is not even clear to me if only tropospheric aerosols were selected (maybe I missed reading it somewhere). I understand that the authors comment on this in Section 4.5, but I would really appreciate if the authors do a quick sensitivity study (e.g. maybe over one of the hot-spots) by incorporating realistic vertical distribution of aerosol and cloud layers, to be able to get an idea of the uncertainty.

---

## Author Comment (AC2) · 15 Mar 2019

We thank Dr Devasthale for his kind words and thoughtful comments. We are pleased to announce that the manuscript will be shorter after deleting section 3.2.2, 3.2.3, appendix B4 and all of its dependencies (following one of reviewer #3's comments).

We particularly appreciate Dr Devasthale's comment on the impact of assumed aerosol and cloud vertical distribution on DARE_OWC. His suggestion has led us to substantially re-write and improve section 4.5. ("assuming fixed aerosol and cloud vertical

layers") to add more discussion of previous work on the subject. We are very grateful as we think this improves this section (and therefore our paper). As written in this section, multiple peer reviewed papers have emphasized the minimal impact of the height of the aerosols above clouds in the calculation of DARE_OWCs, as compared to the effect of changes in other parameters such as the AOD, SSA, or cloud albedo. For this reason we have not included any further sensitivity analysis varying the aerosol and cloud height in our calculations in the present work.

This is how section 4.5 reads now:

"Finally, Long Wave (LW) radiative forcing is particularly dependent on the vertical distribution of aerosols, especially for light absorbing aerosols [Chin et al., 2009]. This is because the energy these aerosols reradiate depends on the temperature, and hence their altitude. For example, Penner et al. [2003] emphasize the importance of soot and smoke aerosol injection height in LW TOA DAREall-sky (see Eq. 1) simulations (higher injection heights tend to enhance the negative LW radiative forcing). Quijano et al. [2000], Chung et al. [2005] and Chin et al. [2009] demonstrate the importance of an aerosol height, in relation to a cloud height (i.e., the aerosols located above, within or below the clouds) in an accurate estimation of SW TOA DAREall-sky. Chung et al. [2005], for example, show that varying the relative vertical distribution of aerosols and clouds leads to a range of global anthropogenic SW TOA DAREall-sky from -0.1 to -0.6 Wâÿśm-2 (using a combination of MODIS satellite, AERONET ground-based observations and CTM simulations, see their Table 2). However, here, we concentrate on cases of aerosol layers overlying clouds in order to compute SW TOA DAREcloudy. Aerosol and cloud layer heights are assumed constant over the globe in our study (see section 2.2). Future studies should incorporate mean gridded (i.e., 4°x5° in this study)-seasonal CALIOP Level 2 aerosol and cloud vertical profiles into the calculation of DAREOWC. However, constraining clouds between 2 and 3km in our study does not seem unreasonable as our AAC AOD calculations using the DR method can only be applied to aerosols overlying specific low opaque water clouds with, among other criteria, an altitude below 3km (see Table B2). On the other hand, constraining aerosols between 3 and 4km in our study is not realistic over many parts of the globe (e.g., see Fig. 7 of Devasthale et al. [2011]). For example, over the region of South East Atlantic during the ORACLES campaign, the HSRL team observed an aerosol layer located in average between 2 and 5km, and overlying a cloud at an average altitude of 1.2km. According to Zarzycki et al. [2010], the underlying cloud properties are orders of magnitude more crucial to the computation of DAREcloudy than the location of the aerosol layer relative to the cloud, as long as the aerosol is above the cloud. In other words, the forcing does not seem to depend on the height of the aerosols above clouds as much as other parameters such as the AOD, SSA or cloud albedo. Zarzycki et al. [2010] investigated this assumption and found that over low and middle clouds, forcing changed by $\sim$1-3% through the heights where the Black Carbon burden was the largest. These small changes in forcing are likely products of a change in atmospheric transmission above the aerosol layer [Haywood and Ramaswamy, 1998] (e.g., a change in the aerosol height linked to a change in the integrated column water vapor above the aerosol layer and this, in turn, would alter the incident solar radiation)."

---

## Author Response (AR1)

Our answers to reviewers are written in blue

**Anonymous Referee #3**

This paper presents a new estimate of the shortwave direct radiative effect attributable to aerosols above
clouds. The paper builds on the previous literature by advancing a technique that applies globally and
utilizes the depolarization ratio method applied to global CALIOP observations. Use of the
depolarization ratio method improves upon a widespread underestimate of aerosol optical thicknesses in
the standard CALIOP retrieval products, but seems to struggle to capture cases of dust over clouds. The
paper does a nice job of summarizing the studies that have come before, the variety of methods that
have been applied to this problem, and the problems that hinder the precise quantification of the global
radiative effect. The paper also does a nice job of placing their quantitative results in the context of
other estimates. I have only some minor comments. After addressing these, the paper should be suitable
for publication in ACP.

We thank the referee for their kind remarks and their very thorough reading of our lengthy manuscript.

1.a. *AR#3:* The analysis is restricted to clouds that are determined to be opaque, but the method by
which opaque clouds are distinguished from clouds that are not opaque is not clear. In the appendix it is
noted that the "CALIOP opacity flag" is used. There should be a brief mention in the body of the paper
of the physical basis for the "opacity flag".

*Our answer:* We now include this short description of the physical basis for the CALIOP opacity flag at
the beginning of section 2.1:

*"Because the CALIOP backscatter signal is totally attenuated below the lowest "feature" detected*
*within any profile [Vaughan et al., 2009], this lowest feature is defined as being opaque.*
*Approximately 69% of the time, the opaque feature detected in a profile is the Earth's surface [Guzman*
*et al., 2017]. In the remainder of the cases, the opaque feature is either a water cloud, an ice cloud, or,*
*very rarely, an aerosol layer." (…) "(1) only one cloud can be detected within a 5 km (15 shot) along-*
*track average (…) and (2) this one cloud must be opaque (i.e., lowest feature detected in a column, and*
*not subsequently classified as a surface return)."*

1.b. *AR#3:* Are there particular regimes where low clouds are prevalent but transparent?

*Our answer:* Yes. As shown by Figure 5 in Leahy et al. [2012], while transparent low clouds occur
globally, they are much more prevalent in the southern oceans and, to a lesser extent, in the northern
Pacific.

1.c. *AR#3:* There is a vague reference to "clouds such as the ones reported in Leahy et al. (2012)". A more specific description would be better.

*Our answer:* Our revised description of the Leahy reference now reads as follows in section 2.1:

*"However, because the DR retrieval technique requires backscatter measurements from opaque water clouds [Hu et al., 2007b], it cannot be used to retrieve AOD from aerosols lying above the low, transparent water clouds that are frequently observed over remote oceans, especially in the southern hemisphere (e.g., Leahy et al. [2012]; Mace and Protat [2018]; O et al. [2018])."*

2. *AR#3:* Figure 1 indicates that the Southern Ocean is the most prominent place on the globe for uniform single layer clouds, but panel b suggests that they are not suitable for the depolarization ratio method. Perhaps it is not of great importance if most of this region has little appreciable aerosol above the cloud layer. Nevertheless, I was left wondering why. Is it a quality of the clouds? Or merely a lack of aerosol optical thickness?

*Our answer:* Please see our response to the previous comment. The issue is the quality of the clouds; i.e., the clouds in the Southern Ocean are often transparent, and transparent clouds are not suitable for analysis using the DR method. We hope that the additional references (i.e., *Mace and Protat [2018]; O et al. [2018]*) will provide some further insights into the nature and causes of these geometrically and optically thin clouds.

3. *AR#3:* Panel d of figure 1 shows a substantial underestimate of cases of aerosol above cloud compared to a similar statistic based on the standard CALIOP aerosol optical thickness product on continents and for oceanic regions dominated by dust plumes. This is discussed in a couple of places in the manuscript, but nevertheless I remained confused as to the cause. The only indication in the body of the paper on line 316 where it says ". . .filtering out of 'unobstructed' but potentially aerosol-contaminated OWCs." The paper does not make clear what "obstructed" or "unobstructed" means in this context or why such clouds would be filtered. This sentence is in dire need of some plain English.

*Our answer:* In response to the referee's remark, we made numerous changes to the text in this paragraph. In particular, we replaced this sentence: *"One reason for the lack of AAC cases offshore from the west coast of Africa in our dataset is the filtering out of "unobstructed" but potentially aerosol-contaminated OWCs (see section B3 in the appendix for more details)"* with this more in-depth explanation: *"The lack of AAC cases offshore from the southwest coast of Africa in the DR method dataset is the result of our conservative data filtering strategy. Because the IABs of aerosol-contaminated OWCs can differ significantly from those measured in pristine, aerosol-free conditions, OWCs suspected of being aerosol-contaminated (which are ubiquitous in this part of the world and very common over continents) are specifically excluded from our DR method analyses (see appendix section B3 for more details)."*

4. *AR#3:* Another place where the description is so technical as to hide the point is in the discussion of the extinction-to-backscatter ratios in sections 3.2.2 and 3.2.3. My sense is that there is an important point in these sections and that differences in the probability distributions in figure 6 must be significant. But it was not clear what that point is or what the significance to the main result of the paper
is.

*Our answer:* This comment was particularly helpful to us. Thank you. The article under review is the
result of many years of analysis. There was a time when this work was separated in two parts
describing, on the one hand, our AAC aerosol optical depths (AOD) paired with CALIOP AAC
extinction-to-backscatter values (S_AAC) and, on the other, the Direct Aerosol Radiative Effects above
clouds (DARE_cloudy). The S_AAC values were there to illustrate the different aerosol types present
above clouds.
Our ultimate goal in this paper now being the calculation of global DARE_cloudy, and knowing that
S_AAC values are not needed in our calculation of DARE_cloudy, these S_AAC are more of a
distraction to the reader. As a consequence, we have deleted section 3.2.2, 3.2.3, appendix B4 and all of
its dependencies. We plan to publish these results separately.

5. *AR#3:* Minor point: In the sentence beginning in line 308 the authors state ". . .negative (positive)
values in blue (red) show the number of AAC cases that are missed (gained). . ." Way back in 2010
Prof. Robock pleaded with us to end this misuse of parentheses [Robock, A. (2010), Parentheses are
(are not) for references and clarification (saving space), Eos Trans. AGU, 91(45), 419–419,
doi:10.1029/2010EO450004]. My understanding is that one of the publishers in our field has
specifically written it out of their style guide. I read pretty widely and the only genre of writing where I
have experienced this application of parentheses is in the atmospheric sciences journals. I hope the
authors will consider rewriting this sentence.

*Our answer:* We have re-written the sentence. Many thanks for the Robock reference.

**Referee #2 Dr Abhay Devasthale**

I will keep the review short and to the point. If not the lengthiest, it is one of the lengthiest manuscripts
I have reviewed so far. So it took me some time to go through it few times and come to the grips of how
the DARE_OWCs are actually computed. But once I stated reading it carefully, it was easier to follow
and understand. I appreciate the hidden efforts behind the work needed to bring onboard information
from the suit of sensors. I also appreciate the way authors contrast and compare their results with the
previous studies. Table 5 is a good idea and could be useful for evaluating models. As far as the
methodology and results are concerned, I do not see anything that should raise a red flag.

*Our answer:* We thank Dr Devasthale for his kind words and thoughtful comments. We are pleased to
announce that the manuscript will be shorter after deleting section 3.2.2, 3.2.3, appendix B4 and all of
its dependencies (following one of reviewer #3's comments).

I do however have one key concern as mentioned below. CALIOP offers two distinct advantages over
passive sensors, namely its superiority in detecting aerosol layers and their precise altitudes. While the
authors go to such a great length and detail to be as realistic and up-to-date in taking into account
aerosol and cloud layers (and their properties) as possible, if I am not mistaken, the altitude of these
layers is assumed to be constant globally. And I can't help but wonder how this is going to affect their
estimates, given the diversity in the verticality of aerosol and clouds in the AAC scenarios and its
impact on DARE_OWCs. It is not even clear to me if only tropospheric aerosols were selected (maybe I
missed reading it somewhere). I understand that the authors comment on this in Section 4.5, but I would
really appreciate if the authors do a quick sensitivity study (e.g. maybe over one of the hot-spots) by
incorporating realistic vertical distribution of aerosol and cloud layers, to be able to get an idea of the
uncertainty.
We particularly appreciate Dr Devasthale's comment on the impact of assumed aerosol and cloud
vertical distribution on DARE_OWC.  His suggestion has led us to substantially re-write and improve
section 4.5. "assuming fixed aerosol and cloud vertical layers" to add more discussion of previous work
on the subject. We are very grateful as we think this improves this section (and therefore our paper). As
written in this section, multiple peer reviewed papers have emphasized the minimal impact of the height
of the aerosols above clouds in the calculation of DARE_OWCs, as compared to the effect of changes
in other parameters such as the AOD, SSA, or cloud albedo. For this reason we have not included any
further sensitivity analysis varying the aerosol and cloud height in our calculations in the present work.
This is how section 4.5 reads now:

[revised manuscript text omitted]

**Page 11: [1] Deleted**    **Meloë Kacenelenbogen**    **3/6/19 10:02:00 AM**

**Page 34: [2] Deleted**    **Meloë Kacenelenbogen**    **3/6/19 9:34:00 AM**

**Page 76: [3] Deleted**    **Meloë Kacenelenbogen**    **3/6/19 10:00:00 AM**